# A highly dynamic F-actin network regulates transport and recycling of micronemes in *Toxoplasma gondii* vacuoles

Javier Periz[1], Mario Del Rosario [1], Alexandra McStea [2], Simon Gras [3], Colin Loney [4], Lin Wang [2], Marisa L. Martin-Fernandez[2] & Markus Meissner [1,3]

The obligate intracellular parasite *Toxoplasma gondii* replicates in an unusual process, described as internal budding. Multiple dausghter parasites are formed sequentially within a single mother cell, requiring replication and distribution of essential organelles such as micronemes. These organelles are thought to be formed de novo in the developing daughter cells. Using dual labelling of a microneme protein MIC2 and super-resolution microscopy, we show that micronemes are recycled from the mother to the forming daughter parasites using a highly dynamic F-actin network. While this recycling pathway is F-actin dependent, de novo synthesis of micronemes appears to be F-actin independent. The F-actin network connects individual parasites, supports long, multidirectional vesicular transport, and regulates transport, density and localisation of micronemal vesicles. The residual body acts as a storage and sorting station for these organelles. Our data describe an F-actin dependent mechanism in apicomplexans for transport and recycling of maternal organelles during intracellular development.

[1] Wellcome Centre for Integrative Parasitology, Institute of Infection, Immunity & Inflammation, University of Glasgow, Glasgow, UK. [2] Central Laser Facility, Research Complex at Harwell Science & Technology Facilities Council, Harwell Campus, Didcot, UK. [3] Experimental Parasitology, Department for Veterinary Sciences, Ludwig-Maximilians-University Munich, Munich, Germany. [4] MRC-University of Glasgow Centre for Virus Research, Sir Michael Stoker Building, Garscube Campus, Glasgow, UK. Correspondence and requests for materials should be addressed to J.P. (email: Javier.Periz@glasgow.ac.uk) or to M.M. (email: Markus.Meissner@lmu.de)

*T*oxoplasma gondii is a member of the Apicomplexa. It infects one third of the human population and can cause serious disease in pregnant women and immunocompromised individuals. Furthermore, psychiatric diseases might be associated with chronic infection, since the parasite establishes tissue cysts, mainly in the brain[1].

Apicomplexan parasites replicate by internal budding, where multiple daughter cells are formed within the mother[2]. Depending on the number of daughter parasites formed within the mother after each replicative cycle, it is called endodyogeny (two daughter cells), endopolygeny (multiple rounds of mitosis without nuclear division—the resulting polyploid nucleus is later divided into daughter parasites during budding) and schizogony (multiple rounds of mitosis followed by synchronous budding)[2].

In the case of T.gondii tachyzoites, only two daughter cells are formed in a highly synchronised manner in a process called endodyogeny. Many aspects of daughter cell assembly within the mother cell have been described using transmission electron microscopy[3] and more recently using time lapse analysis with different fluorescent organellar markers[4,5]. However, the molecular mechanisms involved in the assembly and regulation of organellar biogenesis in the daughters as well as the disassembly in the mother are not well understood.

While the endosymbiotic organelles, a single mitochondrion and apicoplast (a chloroplast-like organelle, derived by secondary endosymbiosis) divide and segregate into the forming daughters tightly coupled to the cell cycle[6], biogenesis of the unique secretory organelles and the Inner Membrane Complex (IMC, a Golgi derived organelle of flattened membrane vesicles that are tightly associated with the subpellicular microtubules and localised beneath the plasma membrane) is tightly linked to the secretory pathway of the parasite[7,8]. The secretory organelles (micronemes, rhoptries and dense granules) are thought to be formed de novo in the developing daughters and disassembled in the mother.

Although some trafficking factors involved in the biogenesis of these organelles have been identified[7], the exact molecular mechanisms underlying vesicular transport are unknown. Based on previous studies using inhibitors, it has been assumed that daughter cell assembly is driven by microtubule-based mechanisms, since treatment of parasites with microtubule disrupting drugs leads to a disruption of daughter cell assembly[9], while treatment with actin disrupting drugs such as cytochalasin D (CD) did not cause severe defects, apart from an enlarged residual body (RB) that contained maternal organelles. Recent findings demonstrated that parasites form an extensive nanotubular network in a F-actin[10] and myosin[11] dependent manner that is required for material exchange between individual parasites. This raises the possibility that actin-based vesicular transport mechanisms are involved in daughter cell assembly and recycling of maternal material from the mother to the daughter parasites.

Apicomplexan F-actin has been mainly investigated in the context of its important role in gliding motility and host cell invasion, where it was thought to act between the plasma membrane and the IMC of the parasite[12], but recently, multiple functions of F-actin during the life cycle of the parasite have been identified. Using a conditional mutagenesis approach based on DiCre-recombinase we demonstrated that parasite actin is essential for maintenance of the apicoplast[13,14] and maturation of the parasite[14]. Furthermore, depletion of the unconventional myosin, MyoF, results in a similar phenotype, with parasites losing the apicoplast[15,16]. The role of parasite F-actin in apicoplast inheritance appears to be conserved, since disruption of *Pfact1* in *Plasmodium falciparum* results in loss of the apicoplast[17].

While these studies suggested that parasite actin is involved in critical intracellular functions, the mechanisms involved remained obscure due to the inability to detect and visualise F-actin dynamics in the parasite. We recently adopted a new approach for the imaging of F-actin in parasites[10] that is based on the expression of actin-chromobodies (Cb). Visualisation of F-actin in T. gondii using Cbs demonstrated that individual parasites within the PV are connected via an extensive intravacuolar network that appears to be critical for the organisation of parasites within the PV, for regulation of parasite replication and material exchange between parasites[10]. In good agreement, conditional mutagenesis of *act1* results in asynchronous replication, aberrant parasite organisation, failure of parasites to mature, and a block of coordinated parasite egress[10].

Based on these findings, we hypothesised that actin-based vesicular transport mechanisms are critical for daughter cell assembly and recycling of maternal material from the mother to the daughter parasites.

Here we analyse the fate of maternal organelles during parasite replication using MIC2, a well characterised microneme adhesin[18] with a role in attachment and parasite egress[19] as a marker for micronemal organelles. We demonstrate that micronemes are efficiently recycled from the mother parasites to the forming daughters. This recycling process depends on parasite F-actin dynamics, while de novo synthesis appears to be F-actin independent.

Analysis of vesicular transport along F-actin demonstrates that it occurs within the cytosol of the parasite, is multidirectional, and follows the direction of actin flow. Finally, our data demonstrate that the RB acts as a storage and sorting station to redirect vesicle transport of maternal material, supporting a general mechanism of transport associated with actin.

## Results

**Maternal MIC2 vesicles are recycled during replication**. During the late stages of parasite replication, micronemes are formed de novo in the daughter cells, while the micronemes of the mother parasite disappear, which led to the hypothesis that they are simply disassembled[6]. We hypothesised that micronemes of the mother are recycled to the daughters, meaning that each daughter should obtain ~50% of the mother's micronemes and is required to form the other 50% de novo. Furthermore, after each replicative cycle (parasites replicate every ~6 h), recycled material should be evenly distributed among daughter parasites.

To test this hypothesis, we followed the fate of maternal organelles during endodyogeny, focusing on the parasites micronemes. We followed the trafficking, de novo synthesis and recycling of micronemal proteins during endodyogeny using a dual-labelling strategy (Fig. 1a). Therefore, micronemal protein 2 (MIC2), a well characterised adhesin[18] was tagged with a multifunctional reporter HaloTag[20] (hereafter, MIC2-Halo) in RHdeltaKu80[21] parasites. We ensured functionality of MIC2-Halo by verifying its correct location in the micronemes (Fig. 1) and normal behaviour during gliding motility of the parasite, since it is secreted and found in trails, indicating proper processing by rhomboid proteases[22,23] (Supplementary Fig. 1).

We first labelled MIC2-Halo of the mother cell with TMR dye for 48 h, before parasites were extracted and allowed to invade and replicate in HFF cells. Subsequently, de novo synthesised MIC2 was labelled with SiR dye, resulting in dual labelling of old (=recycled) and de novo synthesised MIC2-Halo (Fig. 1a, b). (Fig. 1a, b). Using structured illumination microscopy (SIM), we detected two well resolved, independent subsets of micronemes containing either recycled or de novo MIC2-Halo. Recycled MIC2-Halo was evenly distributed among the daughter cells,

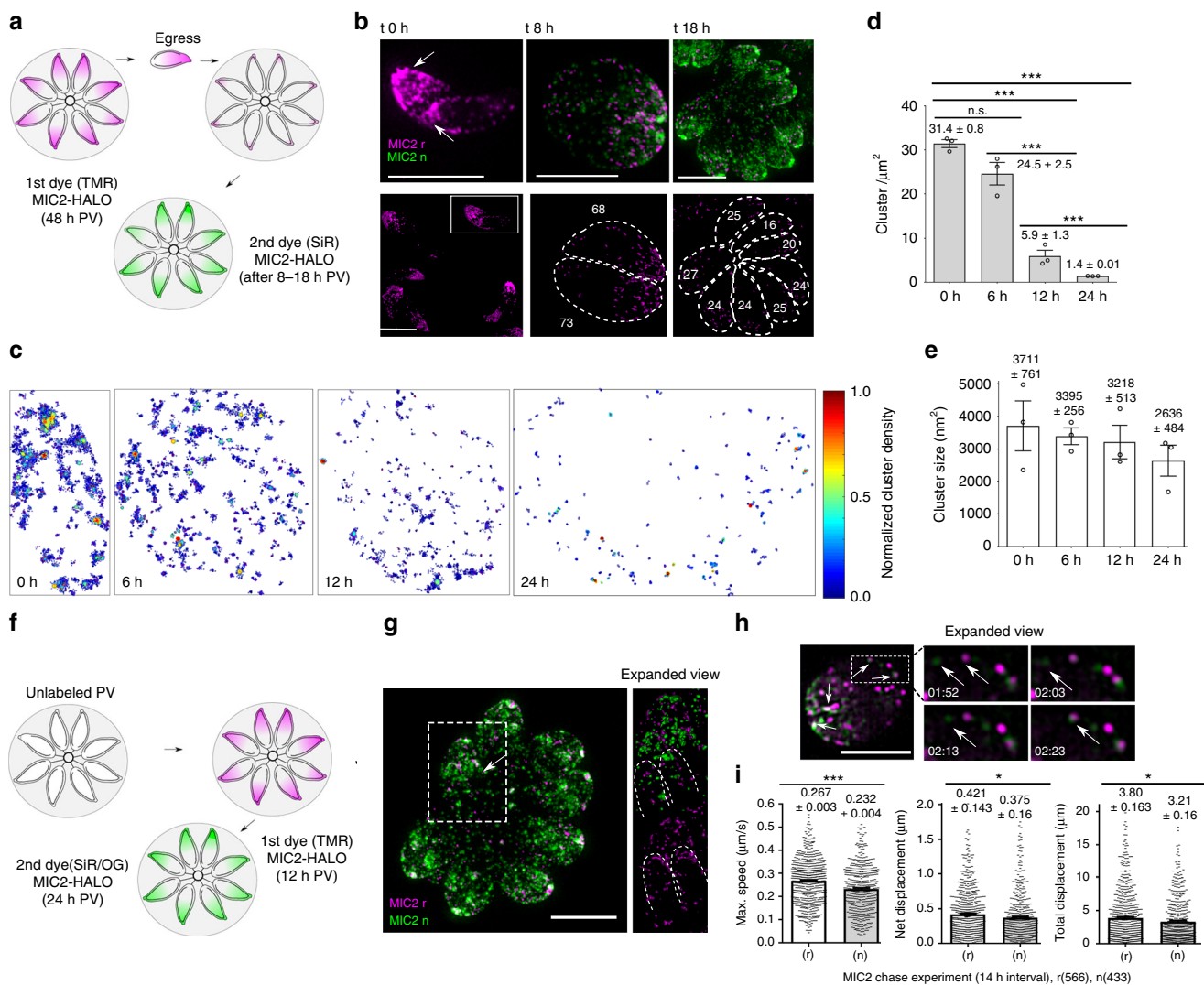

**Fig. 1** Maternal MIC2 is recycled during parasite replication. **a** Schematic illustrating dual labelling strategy to monitor vesicle recycling. Maternal MIC2 was labelled with TMR and de novo synthesised MIC2 was labelled with Silicon-rhodamine (**b**, **g**) or Oregon green (**h**, **i**). **b** Top panel. 3D-SIM images of parasites containing recycled and de novo MIC2 8 h and 18 h post invasion (white arrows show individual cells). Bottom panel shows distribution and recycling of vesicles from a single cell stage to two and eight PV stages (estimated of numbers in each cell are shown in white). **c** Molecular density maps of MIC2 vesicle clusters from representative PVs taken at 0–6–12–24 h. Relative density is normalised and represented with a pseudo-coloured scale. **d** MIC2 cluster density from representative data shown in **c**. **e** Estimated size of MIC2 vesicle clusters from representative data shown in **c**, **d**. Data shown in **c**, **d** and **e** correspond to vacuole triplicates n = 3 of each condition measured and analysed using SMLM, DBSCAN and one way ANOVA, (Tukey's test). At least 20,000 frames were analysed per measurement. **f** Schematic illustrating a pulse-chase experiment in PVs using a dual labelling strategy. PVs expressing MIC2-HALO were sequentially labelled after 12 and 24 h with indicated fluorescent HALO ligands. **g** Representative 3D-SIM image shows a 8 stage PV with differential spatial distribution of recycled (magenta) and de novo (green) MIC2. **h** Representative SIM movie frames from PVs show association (white arrows) of recycled (r) MIC2 vesicles (magenta) and vesicles synthesised de novo (n, in green). Note that no fusion of the two subpopulations occurs. (Time scale in frames are shown in minutes). **i** Quantification of vesicles trajectories from vesicles recycled (r) and synthesised de novo (n) Two stage PVs were labelled within a 14 h interval. Data were analysed with icy automatic particle tracking, n = 566 vesicles (r) and n = 433 (n) de novo vesicles from triplicate 14 h PVs compared with two-tailed unpaired t tests. Error bars indicate mean ± SEM. ***p < 0.0001, **p < 0.01, *p < 0.05. Scale bars, 5 μm. Source data are provided as a Source Data file

demonstrating that maternal micronemes are efficiently recycled into the daughters and not, as previously assumed, disassembled at the final stage of replication (Fig. 1b).

Our observations show that we can distinguish, unambiguously, MIC vesicles formed de novo from those recycled from the mother based on the following evidence. Firstly, the localisation pattern—since, after each round of replication, de novo synthesised MIC2-Halo and recycled vesicles occupy specific locations—de novo, mainly antero apical localisation, and recycled vesicles, redistributed along the cell. Secondly, the

recycled vesicles from the mother are distributed evenly in daughter parasites after each replication cycle (Fig. 1b).

To characterise and quantify the distribution and organisation of MIC2-Halo containing vesicles during replication in detail, we implement Clus-Doc analysis that combines SMLM (single molecule localisation microscopy)[24,25] and DBSCAN (density-based spatial clustering of applications with noise)[26]. We analysed the distribution of MIC2-vesicles being recycled from the mother to the daughters after 1, 2 and 4 rounds of replication by looking at single cells and PVs after 6, 12 and 24 h (Fig. 1c–e). The results

are fully consistent with the process of vesicular recycling, since the decrease in the cluster density of vesicles per cell (Fig. 1c, d) correlates with parasite replication., The recycling occurs almost quantitatively, and the size of the vesicular clusters did not change significantly during replication (Fig. 1e); this variation in the estimated size of the clusters after 24 h may suggest a level of heterogeneity of vesicles, either due to processing or a different degree of cluster association which cannot be resolved with the type of fluorophore (TMR) used in the study (Fig. 1e).

To investigate differences in localisation and vesicular transport between recycled and de novo synthesised MIC2, PVs were differentially labelled with TMR, and SiR in a 12 h interval (Fig. 1f). De novo synthesised MIC2, is mainly localised and concentrated at the apical end of the replicating daughter cells (Fig. 1g). Pools of vesicles synthesised with a difference of up to 12 h interval are accumulated at the apical end and parasite periphery (white arrows, Fig. 1g), with recycled vesicles being redistributed from the mother into nascent daughter cells. Furthermore, fixed and live imaging using SIM (Fig. 1h, Supplementary Movie 1) demonstrates that both vesicle types, containing de novo or recycled MIC2, can transiently associate with each other along the periphery of the cells. However, the two sub-populations never coalesce, and are transported from the RB into daughter cells and exchanged between individual parasites (Fig. 1h, supplementary Movie 2, left and right panels).

Finally, conventional live imaging using wide field microscopy was used to improve temporal resolution (Fig. 1i). Quantification of the average speed and trajectories of vesicles using automatic tracking show that pools of recycled vesicles (two stage 14 h PV) have slightly faster transport ($0.267 \pm 0.003 \, \mu m \, s^{-1}$ vs. $0.232 \pm 0.004 \, \mu m \, s^{-1}$) and larger trajectories ($0.421 \pm 0.143 \, \mu m \, s^{-1}$ vs. $0.375 \pm 0.16 \, \mu m \, s^{-1}$) than de novo synthesised vesicles (Fig. 1i). Our data suggest a highly dynamic process of vesicle transport from the mother cells to the apical end of the daughter cells. This multi-directional, active transport was observed in posterior areas of the parasite, where no microtubules are present[27], suggesting a microtubule independent transport mechanism.

**Recycled MIC2 vesicles associate with the F-actin network.** Having established that maternal micronemes are recycled during parasite replication, we wished to investigate if F-actin dynamics and the previously identified intravacuolar network[10] is required for material exchange. We expressed F-actin binding nanobodies (Cb) fused to a variety of tags including GFP-emerald (Cb-Emerald FP), photoconvertible mEos3.2[28] (Cb-mEos3.2) and SNAP-tag[29] to visualise MIC2 and F-actin simultaneously (Fig. 2) and performed dual labelling experiments, as described above (Fig. 2a). We observed the same distribution and motility of MIC2-Halo containing vesicles as above, demonstrating that expression of Cb does have no influence on recycling, localisation or transport of micronemes. Importantly, both recycled (r) and de novo (n) MIC2-Halo vesicles are found to decorate a branched network of F-actin bundles interconnecting individual parasites regardless of PV size (Fig. 2b–e). An estimation of the total number of MIC2r and MIC2n indicated that MIC2 vesicles represent two distinct sub-populations: MIC2n can be found mainly at the apical tip of the parasite, while a significant amount of MIC2r appears to be associated with actin bundles localised at the posterior pole (Fig. 2d–g, Supplementary Fig. 2). These vesicles form clusters that bridge distant F-actin bundles and F-actin bundles sometimes "encase" the vesicles suggesting multiple association sites (white arrow in Fig. 2d). The bundles imaged with SIM appear to be variable in diameter and length also suggesting a variability in number of filaments and actin scaffold compaction. Longer F-actin bundles are decorated with vesicles

containing MIC2 and connect the top and bottom of the PV (Fig. 2e and rendered 3D surface z-view). In summary, we obtained unambiguous data from fixed images showing examples of small and big vacuoles with a complex actin network decorated with MIC2-positive vesicles. The estimated ratio of n and r MIC2 vesicles associated to actin bundles in different PV stages, demonstrated a consistent increase of recycled vesicles (r) associated to actin bundles vs. vesicles synthesised de novo (n), which agree with the role of F-actin in the redistribution and recycling of MIC vesicles (Fig. 2f, g).

**Multidirectional transport depends on dynamic and mobile F-actin.** We wished to analyse the dynamics of F-actin bundles associated with MIC2-positive vesicular transport. Therefore, we used live SIM imaging to test whether MIC2 vesicles are associated with active transport on actin filaments.

Using live SIM and conventional wide field imaging (Fig. 3), we confirmed that MIC2-containing vesicles are associated with a highly dynamic F-actin network that connects the posterior end of parasites with the residual body (RB; (Fig. 3a–c, Supplementary Fig. 4, Supplementary Movies 3–5).

F-actin within the cytosol of the parasite and the RB appears to form dynamic bundles that act as tracks guiding redistribution, exchange and recycling of MIC2 vesicles from the posterior end of the parasite to the RB (Fig. 3b, c). As can be best observed in larger PVs, MIC2 vesicles form clusters associated with the F-actin network that converges into the RB (Fig. 3c, d). Clusters of MIC2-vesicles bridge distant F-actin bundles and move on F-actin bundles in close proximity, enabling a mechanism for redistribution of vesicles through the actin network (Fig. 3c, d). Furthermore, the F-actin network is highly mobile and seems to self-associate resulting in the displacement and accumulation of vesicle clusters (stills in Fig. 3e, Supplementary Movie 5, Supplementary Fig. 4, Supplementary Movie 6).

In this context, the residual body appears to function as a dynamic structure working as a sorting station that accumulates and redistributes MIC2-positive vesicles to the daughter cells and within the PV network (Fig. 3c, Supplementary Fig. 3, Supplementary Movies 6–7). Complementary analysis using kymographs show live colocalisation of actin and MIC2-vesicles, and demonstrate how static, anterograde and retrograde actin flow together, with actin self-association measurements closely mirroring those taken for MIC2-vesicular flow (Supplementary Fig. 4).

To analyse the transport kinetics, we limited our recordings to a few minutes to minimise photobleaching and cell stress under illumination with wide field fluorescence and SIM imaging methods (Fig. 3f, Supplementary Movie 5). During this short observation, the majority of the vesicles remain stationary and undergo little displacement. However, in mobile vesicles, transport is long range, multidirectional with lateral, anterograde, retrograde and combined trajectories. Manual tracking enables us to identify variability in the kinetics and trajectories of individual vesicles. MIC2 vesicles exhibited variable speeds (ranging from $0.37$ to $1.27 \, \mu m \, s^{-1}$) and long net displacement of several microns ($5.4 \, \mu m$) which results in exchange of vesicle material between distant daughter cells. Independent kymograph analysis in well resolved actin bundles and vesicles (Fig. 3f, g, particle 5 and 6) show a complex actin flow exhibiting static, retrograde and anterograde directions, suggesting the presence of actin bundles in different dynamic states. Importantly, actin and vesicle flow exhibited similar kinetics, and flow direction (see MIC2 vesicle 5; max. speed $0.3 \, \mu m \, s^{-1}$ and actin flow tracks ranging $0.29–0.53 \, \mu m \, s^{-1}$) and the data closely described the trajectory fluctuations and speed of independent measurements using particle tracking ($0.56–0.43 \, \mu m \, s^{-1}$ Fig. 3f, g).

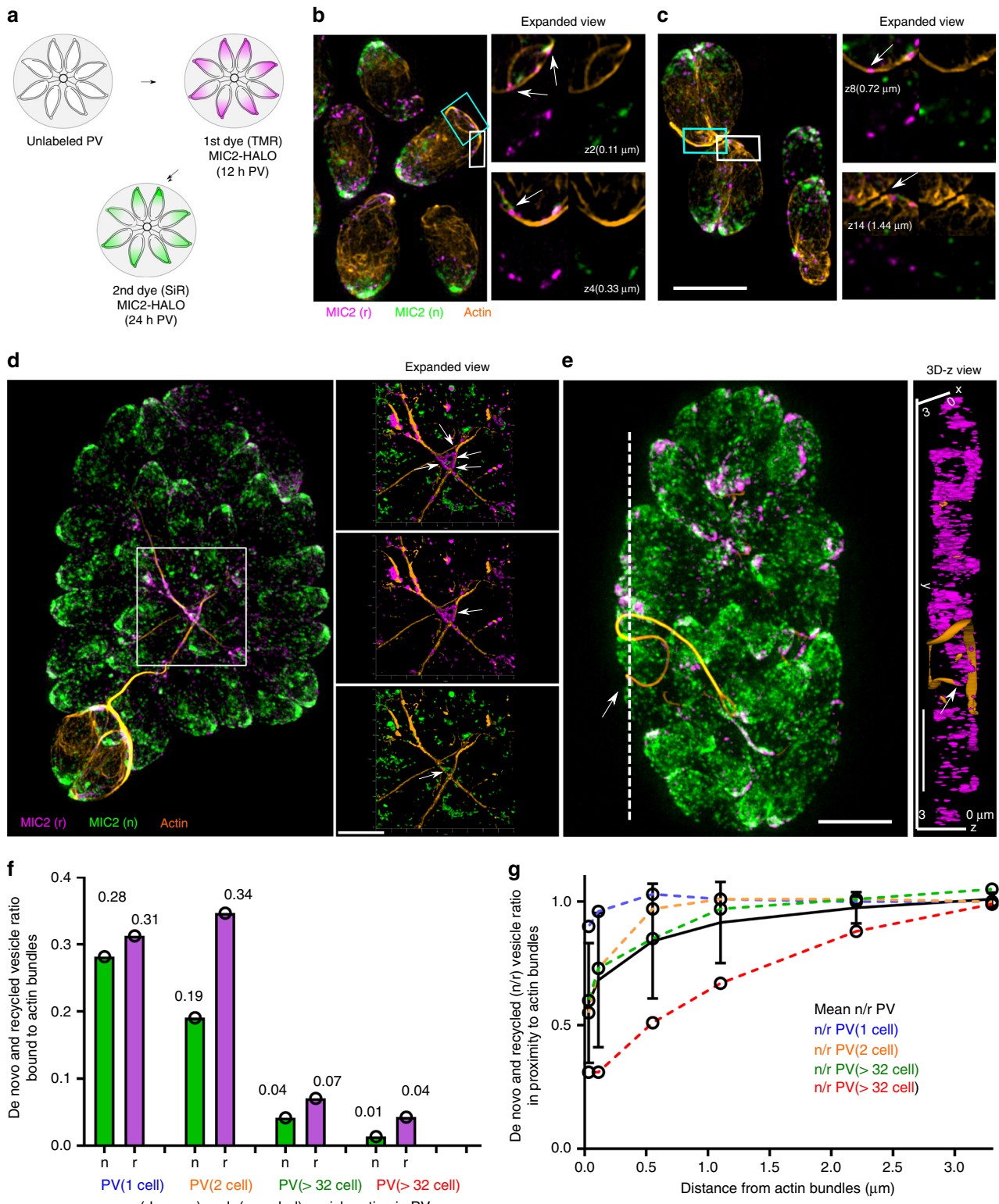

Together these data demonstrate that MIC2-positive vesicles are transported in multiple directions along a highly dynamic F-actin network. Similar transport kinetics for MIC2-containing vesicles has been measured in the parental line, not expressing Cb, demonstrating that Cb has no significant influence on F-actin dynamics (Fig. 1).

To further determine the role of F-actin for vesicular transport, we made use of F-actin-modulating drugs, such as cytochalasin D (CD, disrupts F-actin) or Jasplakinolide (JAS, stabilises F-actin). The association of MIC2 to F-actin is diminished upon treatment with CD and MIC2-positive vesicles localise almost exclusively at the apical periphery of the parasite (Fig. 4a). Under these experimental conditions, MIC2 vesicles appeared densely packed and cannot be resolved using SIM imaging in contrast to untreated PVs (Fig. 3b, c, f) where vesicles are well resolved. When cluster size and colocalisation of MIC2 and actin was

**Fig. 2** MIC2 vesicles clusters associate and bridge actin bundles. **a** Schematic illustrating dual labelling strategy to monitor vesicle recycling. 12 h PVs expressing stably MIC2-HALO and actin-Cb-emerald were sequentially labelled in a 12 h interval. **b**, **c** 3D-SIM representative images of one and two cell stage PVs show MIC2 (r) and (n) vesicles decorating F-actin bundles (yellow) interconnecting the parasites. Expanded view shows z stacks in which the image was taken, and distance in the axial plane relative to the surface. **d** 3D-SIM representative images of large PV showing association of recycled and de novo vesicles with F-actin. Detailed views of 3D-SIM rendered surface images show MIC2 vesicle clusters bridging two independent F-actin bundles (white arrows). **e** 3D-SIM representative images of large PV. Detailed views of 3D-SIM rendered surface images show association of vesicles with the F-actin network connecting the daughter cells. Right: Detailed view of 3D-rendered surface; z view shows recycled MIC2 vesicle clusters associated with the end of the F-filament (white arrow). Scale bars, 5 μm. **f** Quantification of the estimated ratio of n and r MIC2 vesicles associated to actin bundles in different PV stages Data correspond to full field of views (FOVs) from **a** (blue, one cell PV stage; 10 PVs analysed; 1223 n and 521 r number of vesicles), **b** (orange, two cell PV stage, 11 PVs; n 1659, r 896 total number of vesicles), **c** (green, one large PV, n 6695 r 2795 total number of vesicles) and **d** (red, one large PV n 4016 r 713 total number of vesicles). **g** Quantification of the de novo/recycled (n/r) vesicle ratio in proximity to actin bundles in PV stages. Data correspond as in **f** to the full FOVs from **a** (blue), **b** (orange), **c** (green) and **d** (red). Estimated Number of vesicles and ratios of de novo (n) and recycled (r) are calculated at incremental distance to the actin bundle based on a voxel size. (see Methods). Error bars indicate mean ± SDV. Scale bars, 5 μm. Source data are provided as a Source Data file

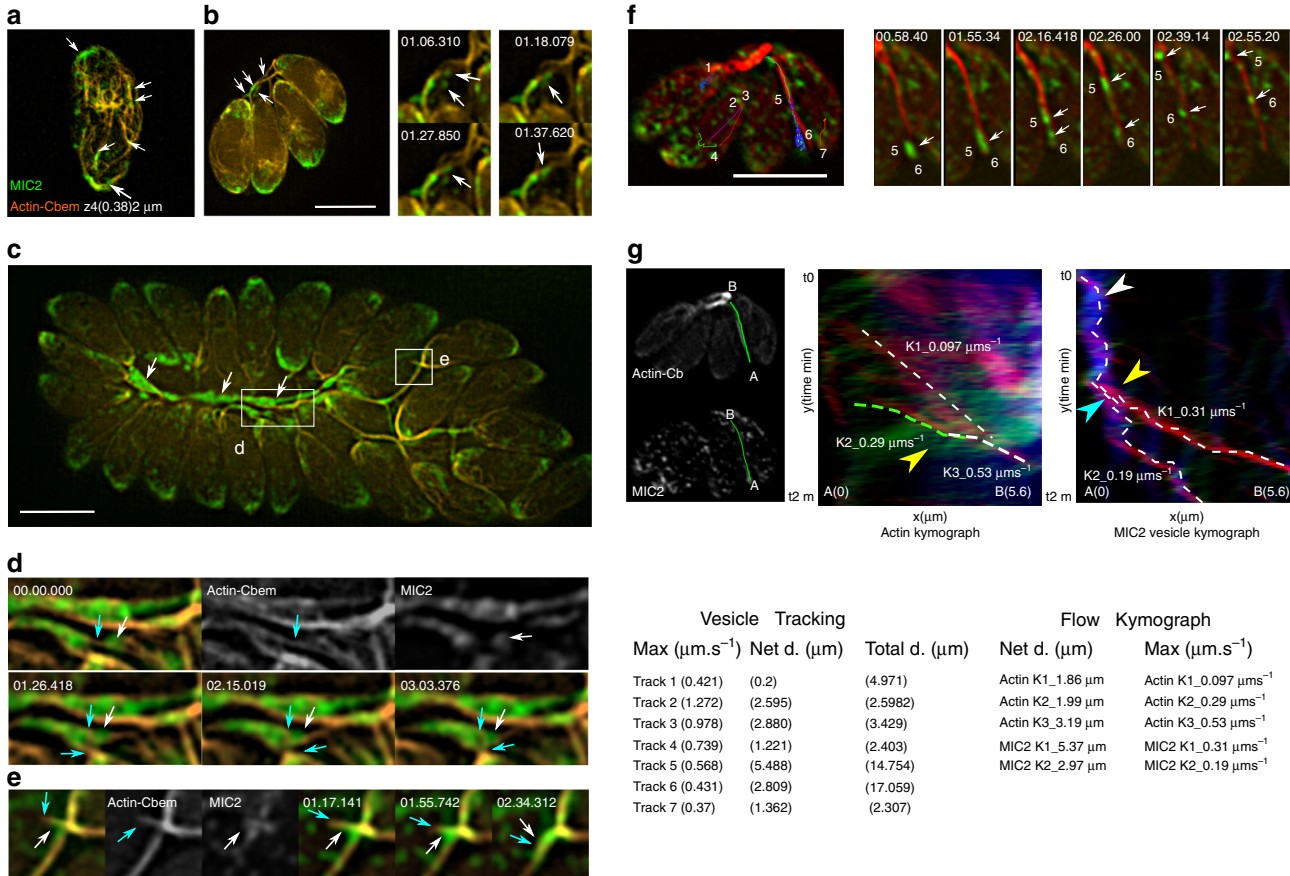

**Fig. 3** Multidirectional transport depends on F-actin dynamics. **a** SIM image showing association of MIC2 vesicles (white arrows) with an F-actin network. (The image corresponds to slice z 4 of a 3D-SIM image, acquired at 380 nm position in the axial plane from the parasite surface). Scale bars, 5 μm, **b** SIM representative movie frames show MIC2 vesicles (white arrows) being transported on an F-actin bundle connecting the parasite posterior ends. Scale bars, 5 μm **c** SIM representative movie frames showing accumulation of MIC2 vesicles in the F-actin network and RB of a large PV. MIC2 vesicle clusters bridge nearby F-actin bundles (square and white arrows). **d** Expanded view from C. MIC2 vesicle transport (square, white arrows) associated with neighbouring actin filaments (cyan arrows). Time scale in frames shown in minutes. **e** Expanded view from C. Fusion of actin filaments (cyan arrows) result in association with nearby MIC2 vesicle populations (white arrows). **f** Wide field representative movie frames, show trajectories and kinetics of MIC2 vesicular transport. Time scale in frames shown in minutes. **g** Top left panels. Kymographs measured Actin and MIC2 vesicle flow. Measurements taken from points A to B. Top central panel. Actin tracks show static, anterograde, retrograde transport in blue, red and green respectively. Yellow arrow shows actin tracks with trajectory and kinetics similar to MIC2 vesicle 5. Top right panel. MIC2 vesicle kymograph. Flow tracks from vesicle 5 and 6. Flow remains static for a large vesicle (blue). After half of the recording time, two flow trajectories with anterograde flow (red) are recorded that correspond with flow kinetics associated with vesicle 5 and 6, respectively. Bottom left panel. Manual tracking of individual vesicles (1–7), displaying anterograde, retrograde, lateral and combined multidirectional trajectories, in the cell and the actin network converging in the RB. Maximum speed, net and total displacement are shown. Number position, indicates the starting time point for each vesicle trajectory. Bottom right panel. Net displacement and maximum speed measurements extracted from kymograph analysis. Error bars indicate mean ± SEM. Source data are provided as a Source Data file

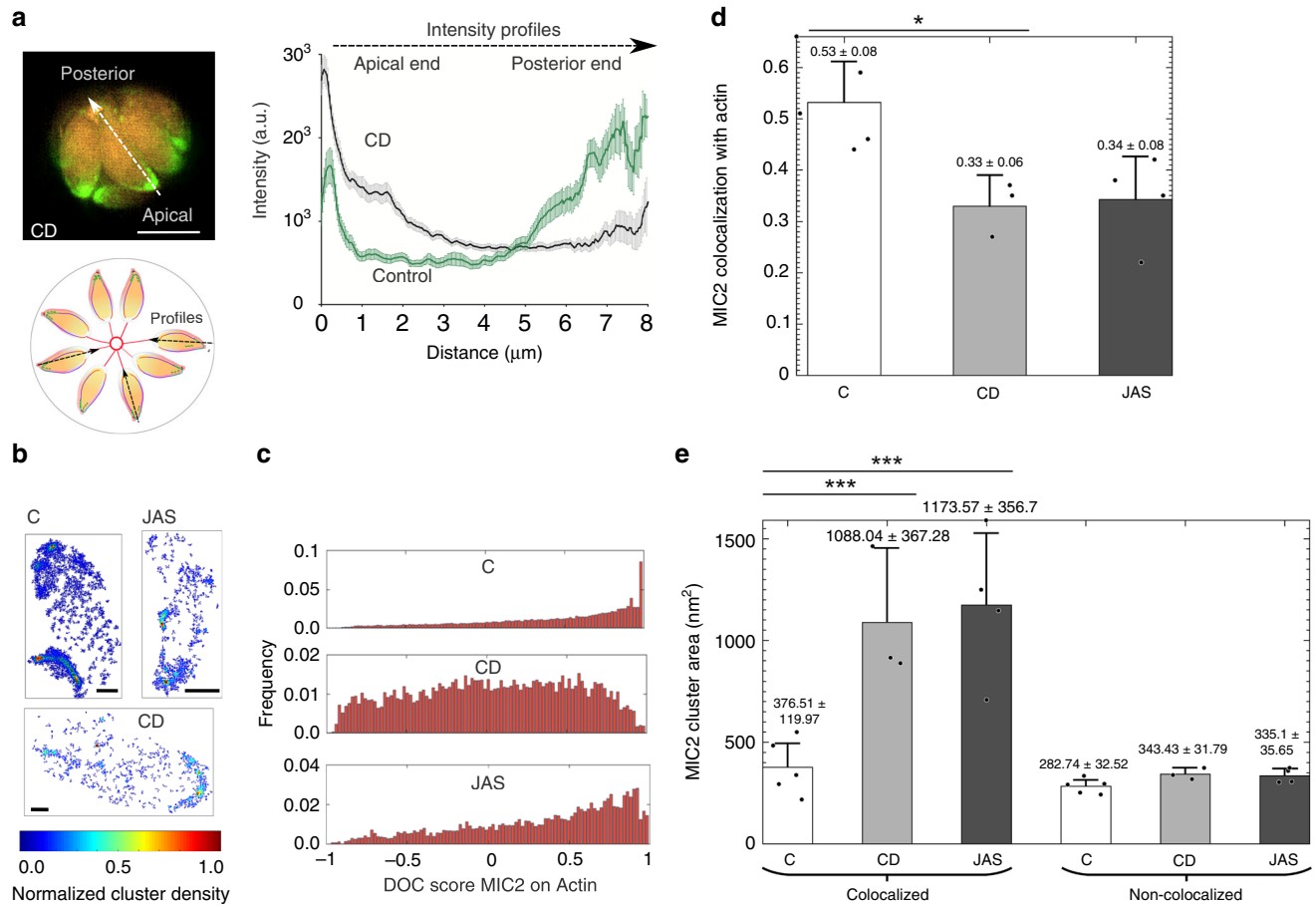

**Fig. 4** Disruption of F-actin dynamics affects microneme recycling. **a** Top left. SIM representative image of four stage PV treated with CD. Bottom left. Schematic showing how the intensity profiles measurements are taken (broken line). Right. Intensity profiles of PVs treated with CD show the disappearance of the actin network and a concentration and re-localisation of MIC2 vesicles in the cell periphery. The graph shows $n = 36$ profiles per condition, corresponding to three measurements per PV, 12 PVs out of 9 independent FOVs. Scale bar 5 μm. **b** Representative ROIs from PVs showing cluster density maps from MIC2 molecules in control (C), JAS, and CD treated PVs. Cluster density maps show a redistribution and loss of MIC2 in the cell treated with drugs affecting the structure of F-actin. The colour scale represents normalized relative density. Scale bar: 1 μm. **c** Frequency distribution of DoC scores of all MIC2 protein co-localised to F-actin. Co-localisation is decreased under CD and JAS treatments that alter F-actin stability. **d** Bar charts of MIC2 colocalisation with actin show a loss of MIC2 co-localisation to F-actin under treatments affecting structure and actin dynamics. **e** Bar charts of MIC2 cluster area show an increase in relative cluster size of MIC2 vesicles associated with actin bundles treated with JAS and CD. Relative average cluster size remains stable in MIC2 vesicle clusters transported in an untreated actin network MIC2 and vesicles non associated to the actin network regardless the treatment condition. Data measured and generated using SMLM, DBSCAN and ClusDoc; MIC2 colocalisation compared with one way ANOVA, (Tukey's test), C corresponds to $n = 5$ PVs analysed; CD $n = 3$; JAS condition $n = 4$ PVs. Error bars indicate mean ± SEM. ***$p < 0.0001$, **$p < 0.01$, *$p < 0.05$. Source data are provided as a Source Data file

analysed using SMLM and Clus-Doc[26] (Fig. 4b–d), treating parasites with CD and JAS (1 h, 37 °C, at concentrations of 1, 200 nM respectively) resulted in a decrease in colocalisation (Fig. 4c, d). Furthermore, the average area of these clusters was significantly increased in comparison to control parasites (Fig. 4e). In contrast, the cluster size of MIC2-vesicles not associated with F-actin remained constant, indicating the existence of two populations of MIC2-vesicles, one associated with F-actin and one not bound to F-actin, which consequently is not affected by F-actin modulating drugs (Fig. 4e).

These results, have mechanistic implications for the role of the actin network in the regulation of vesicular transport. It supports a view, in which changes in structural and dynamic properties of F-actin modify vesicular transport, increase local concentration, resulting in cluster formation. MIC2 vesicles appear to be able to associate, release and re-associate to F-actin bundles, facilitating exchange and communication of vesicles between different actin tracks.

In good support, we found a continuous F-actin network that provides full connectivity between parasites in early replication stages that facilitates recycling and re-positioning of MIC2 vesicles from the cytosolic apical location of the mother cell into the apical end of the daughter cells (Fig. 5a, b Supplementary Movies 7–8, 9a, b).

**F-actin dependent recycling of the IMC**. Having established that maternal micronemes are recycled to the daughter parasites in F-actin dependent manner, we were interested if recycling of the IMC occurs in a similar manner. Previously, it has been shown that the maternal IMC is recycled at the final stage of replication[4] and that disruption of F-actin results in morphological aberrant parasites[14,30], supporting this hypothesis.

We established a parasite line expressing Cb and as marker for the IMC an extra copy of SNAP-tagged MyoA (Fig. 5b, c) and verified that it behaves like control parasites (Supplementary

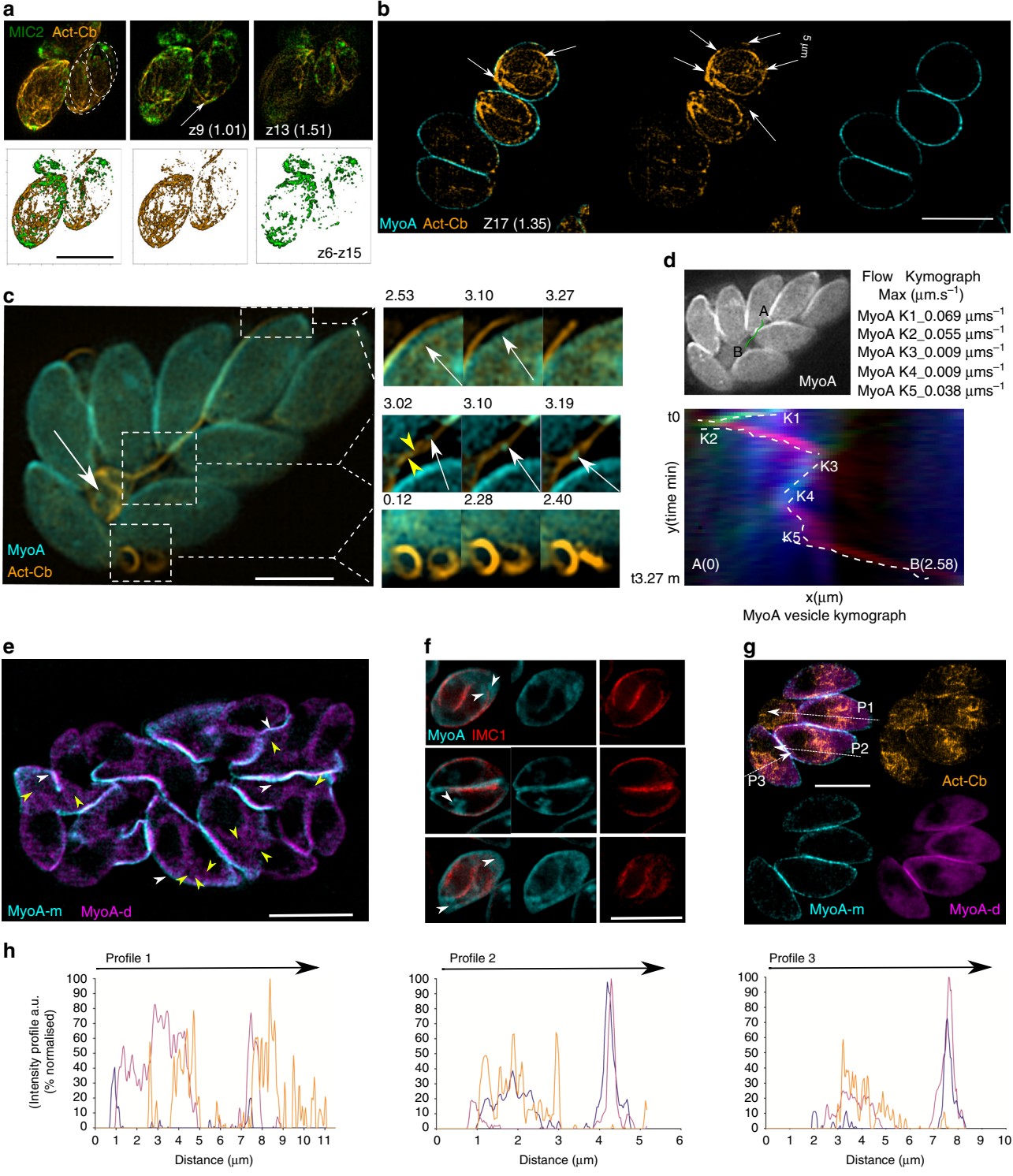

Fig. 5) the core component of the glideosome, which is anchored in between the PM and IMC of the parasite[12]. As expected, the F-actin network connects daughter cells at their posterior ends and is enveloped by MyoA from the mother cell, demonstrating a cytoplasmic location of F-actin during replication (Fig. 5b Supplementary Movie 9a, b). In analogy to the observations made for MIC2-Halo, maternal MyoA can be seen to be transported into the RB (as demonstrated with live and kymograph analysis, Fig. 5c, d) along the intravacuolar F-actin network (Fig. 5c–e; Supplementary Movies 9b, 10), arguing

for a general mechanism for the recycling of maternal IMC material.

When a dual labelling experiment, as described above for MIC2, was performed, to discriminate maternal from de novo synthesised MyoA (Fig. 5e–g), we found that newly synthesised MyoA, appears in a diffuse cytoplasmic localisation and partially co-localises with maternal MyoA (Fig. 5e–g and intensity profiles 5 h), in good agreement with previous results, arguing for a cytosolic formation of the glideosome complex, which subsequently associates with the IMC of the daughter cells at the final

**Fig. 5** A mobile, interconnected F-actin network supports vesicular transport **a** 3D representative SIM image shows a continuous F-actin network connecting mother and daughter cells. Top row. 3D and z slices show continuous F-actin between mother and daughter. Bottom row. 3D surface rendered model and independent channel views comprising z stacks (z6–z15). **b** 3D representative SIM image shows a continuous F-actin network connecting mother and daughter cells underneath MyoA. **c** SIM live imaging stills show MyoA vesicles associated with F-actin bundles located in the RB (left image, white arrow). Detailed views: Detachment of extracytoplasmic F-actin bundles from the parasite surface (top). Recycled MyoA vesicles being transported from the daughter into the RB (centre). Fusion of mobile extra-cytoplasmic circular F-actin bundles (bottom). **d** Kymographs showing the transport of MyoA-vesicle from a daughter cell into the RB. Measurements taken from points A to B. Flow show static, anterograde, retrograde transport in blue, red and green respectively. Estimated kinetics in each tracks are shown. **e** Representative SIM image shows a large PV with MyoA labelled with two different dyes (TMR and SiR) in a 12 h interval to distinguish a pool of maternal MyoA (cyan) at the parasite periphery (white arrow head) from a cytoplasmic and diffuse (yellow arrow head) pool of de novo synthesised MyoA (magenta). **f** Representative SIM image shows a 2 stage PV showing localisation of MyoA (cyan) and IMC1 (red). MyoA shows a less defined localisation as a result of MyoA recycling during parasite replication (white arrows). **g** Representative SIM image shows a 4 stage PV with mother MyoA (cyan) and cytoplasmic, diffused de novo MyoA (magenta) wrapping the actin network. **h** Intensity profiles from figure G indicate a network of actin (orange line) embedded by MyoA synthesised de novo (MyoAd purple line) and from the mother (MyoA m in cyan line). Intensity was expressed as a percentage of the maximum intensity for each marker. Scale bar 5 μm. Z number corresponds to the displayed stack number, brackets show the position of the z stack in the axial plane measured from the parasite surface. Source data are provided as a Source Data file

steps of endodyogeny[31]. In addition, maternal MyoA appears to be distributed and recycled via the intravacuolar network, similar to MIC2 (Fig. 5e, g). We also performed dual labelling of de novo MyoA relative to the IMC-marker IMC1.

At this point, our observations cannot state unambiguously the exact topology of myosin in replicating parasites and how MyoA is transported by cytosolic F-actin, when it is supposed to be located in between the IMC and the PM. We speculate that the IMC may be permeable and MyoA can be located on both the outer and the inner leaflet of the IMC during this replication stage. Together these data demonstrate that the F-actin dynamics is required for recycling of maternal micronemes, IMC and potentially other maternal organelles, such as rhoptries or dense granules.

**F-actin is in association with subpellicular microtubules.** Cluster analysis of MIC2-Halo suggested that one subset is associated with F-actin and reacts to F-actin modulating drugs, whereas a second subset is actin independent (Fig. 4d, e). Furthermore, live imaging demonstrated that many MIC2-positive vesicles move along a highly dynamic F-actin network, but several observations were made, where MIC2-positive vesicles were not or only partially associated with F-actin, especially at the apical pole of the parasite (Figs. 2, 3).

We hypothesised that F-actin and sub-pellicular microtubules might coordinate vesicular transport. Indeed, previous studies have suggested a localisation and interaction of parasite F-actin with subpellicular microtubules[32,33]. We stained the subpellicular microtubules with sir-tubulin[34] in tachyzoites expressing MIC2 and Cb-emerald (Fig. 6a, b).

We performed three colour SIM and found that cytosolic F-actin connects the apical and the posterior end of the parasite (Fig. 6a–c) and is decorated with MIC2 vesicles. Analysis of Z-stacks from 3D-SIM imaging support an arrangement of F-actin in cytosolic localisation (Fig. 6b, c bottom row white arrows and Fig. 6e–g white arrows). This cytosolic actin network connects regions in close proximity to the apicoplast and the trans-golgi network to the parasite periphery. 3D-SIM Z-stacks (Fig. 6h, l together with rendered surface image and Z-stacks) and intensity profiles support a close association of F-actin to microtubules and between microtubules to micronemes.

Although infrequent, other microtubule morphologies are also observed including long microtubule polymers overlapping the length of the parasite (Fig. 6h, k yellow arrows) and short microtubules (yellow arrows in Fig. 6c, j, k) with an apparent size of ~200 nm which are associated with F-actin at the posterior end.

**Transient self-association and mobility of actin bundles.** We identified two types of vesicular transport behaviours which are dependent on F-actin structure; one in which the vesicle translocates on actin tracks, and a second in which the filament bundles are highly mobile and associate transiently with vesicles and nearby bundles, facilitating exchange and clustering of material.

To better understand the structural organisation and the behaviour of actin bundles, we performed live-cell SIM imaging and SMLM. Using live and kymograph analysis, we demonstrate a circular movement of actin bundles associated with anterograde and retrograde flow of actin within the RB. Actin flow resulted in transient association of actin bundles (Fig. 7a, b Supplementary Movie 11) which was complementary and independently investigated with SIM Fig. 6c, d and SMLM imaging (Fig. 7c–e) with a parasite line expressing Cb tagged with mEos3.2[28] (Fig. 7e, f). SMLM showed variable diameter distributions of actin bundles inter-connecting the daughter cells (Fig. 7e, f) and forming a circular actin bundle scaffold in the RB (Fig. 7e). Although the precision of localisation for a molecule is 30 nm average in our SMLM experiments, far beyond the atomic resolution of the diameter of single filaments, our observations support a complex, variable association and compaction of bundles, in which the RB is a result of fast F-actin flow, consisting of short bundles and self-association properties of F-actin in a confined space delimited by the posterior ends of the replicating daughter cells. Overall, these observations agree well with a mechanism in which actin filaments can regulate, exchange and connect MIC vesicles between distant regions of the PV.

## Discussion
Among the most fascinating features of apicomplexan biology is its unique cellular architecture. These parasites evolved a highly complex secretory system (micronemes, rhoptries and dense granules) that is required for crucial roles during the infectious cycle, from gliding motility, host cell invasion, and modification to egress.

The apical end is the site for rhoptry and microneme secretion, with these organelles tightly packed into the anterior portion of the cell. While the anterior of the zoite is focused on invasion, the rest of cell carries the genetic material and tools to grow and develop once in the host cell, including a nucleus and a single mitochondrion, plastid, and Golgi.

The apicomplexan secretory system is highly polarised, consisting of the ER, a single Golgi stack[35] and endosomal-like compartments[36,37]. Furthermore, an elaborate alveolar system, called the Inner Membrane Complex (IMC)[38] and the apicoplast

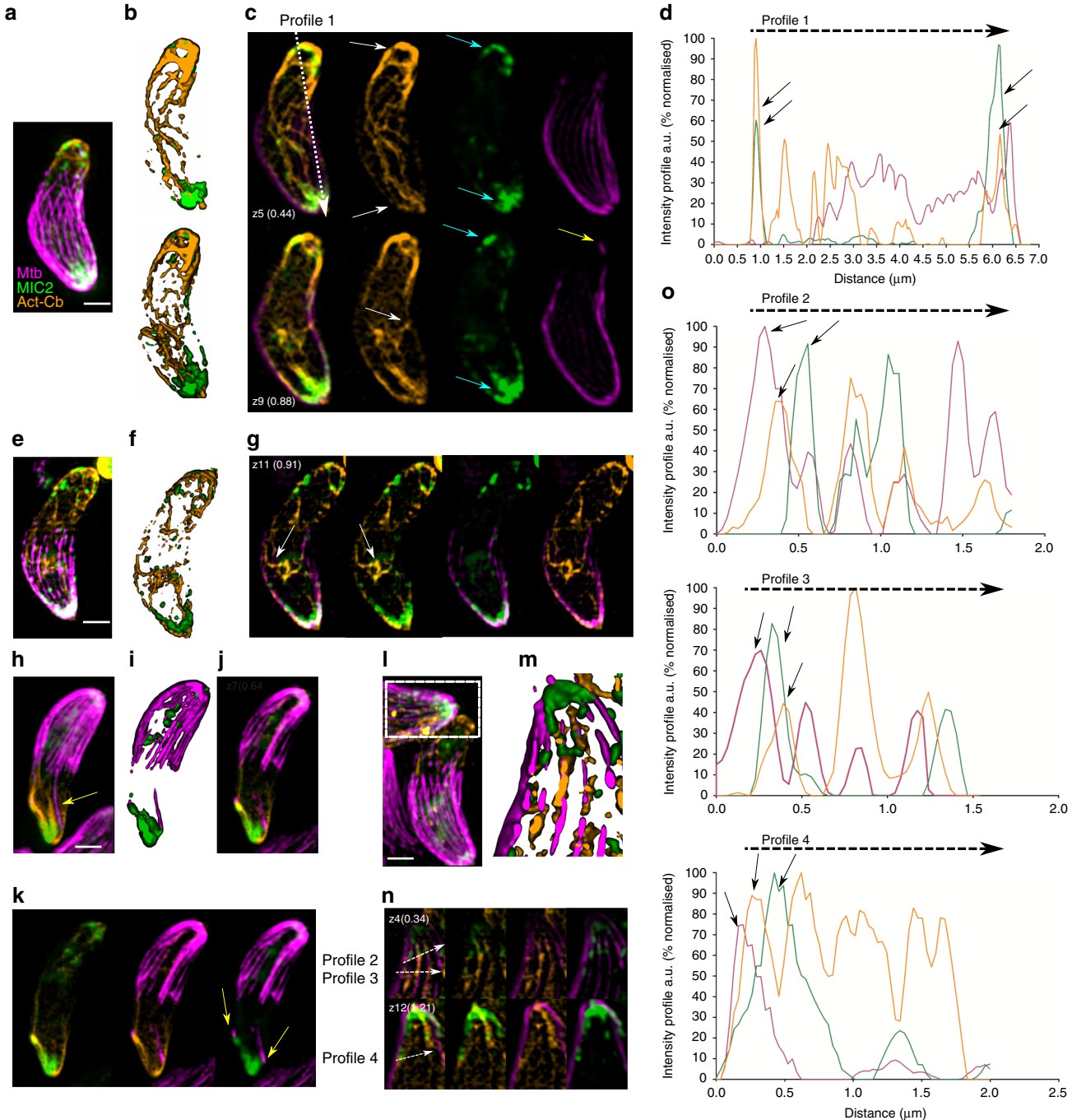

**Fig. 6** F-actin is in association with subpellicular microtubules. **a** Representative 3D-SIM imaging of a tachyzoite shows MIC2 vesicles associated with a continuous F-actin network connecting the apical and the posterior end. **b** Surface rendered 3D images from A). Stacks z1–z5 (top) and z1–z9 (bottom). F-actin network is found connecting MIC2 vesicles to the apical, posterior and lateral sides of the parasite. **c** Z5 and Z9 stacks show overlay and independent channels for actin, MIC2 and microtubules. F-actin bundles accumulate in the apical and posterior end (top panel, white arrows) and are connecting to the lateral periphery of the parasite in close association with microtubules (bottom panel, white arrows). MIC2 vesicles also accumulate and decorate the actin bundle architecture of the posterior and apical ends (cyan arrows). Short microtubules not connected to the long fully structured microtubules are found at the posterior end of the parasite in close association with actin and MIC2 vesicles (yellow arrows). **d** Intensity profiles show unambiguous co-localisation of MIC2 vesicles with actin at the posterior end. Profiles were expressed as a percentage of the highest intensity for each channel. **e**, **f**, **g** 3D surface rendered and Z-stack SIM images of a representative tachyzoite show a continuous, cytosolic F-actin network connecting a region close to the apicoplast and the trans-Golgi network with microtubules in the lateral end of the parasites (white arrows). **h**, **i**, **j**, **k** 3D surface rendered and Z-stack SIM images show a parasite with both short and unusually long microtubules (yellow arrows) localised at the posterior end (yellow arrows). **l**, **m**, **n** 3D surface rendered and Z-stacks SIM images of a representative parasite show microtubules and MIC2 vesicles connected with a cytosolic F-actin network extended in the axial and lateral plane of the parasite. **o** Intensity profiles from N) suggest a connection of a cytosolic F-actin network with microtubules and MIC2 vesicles. Profiles were expressed as a percentage of the highest intensity for each channel. Scale bar 1 μm. Source data are provided as a Source Data file

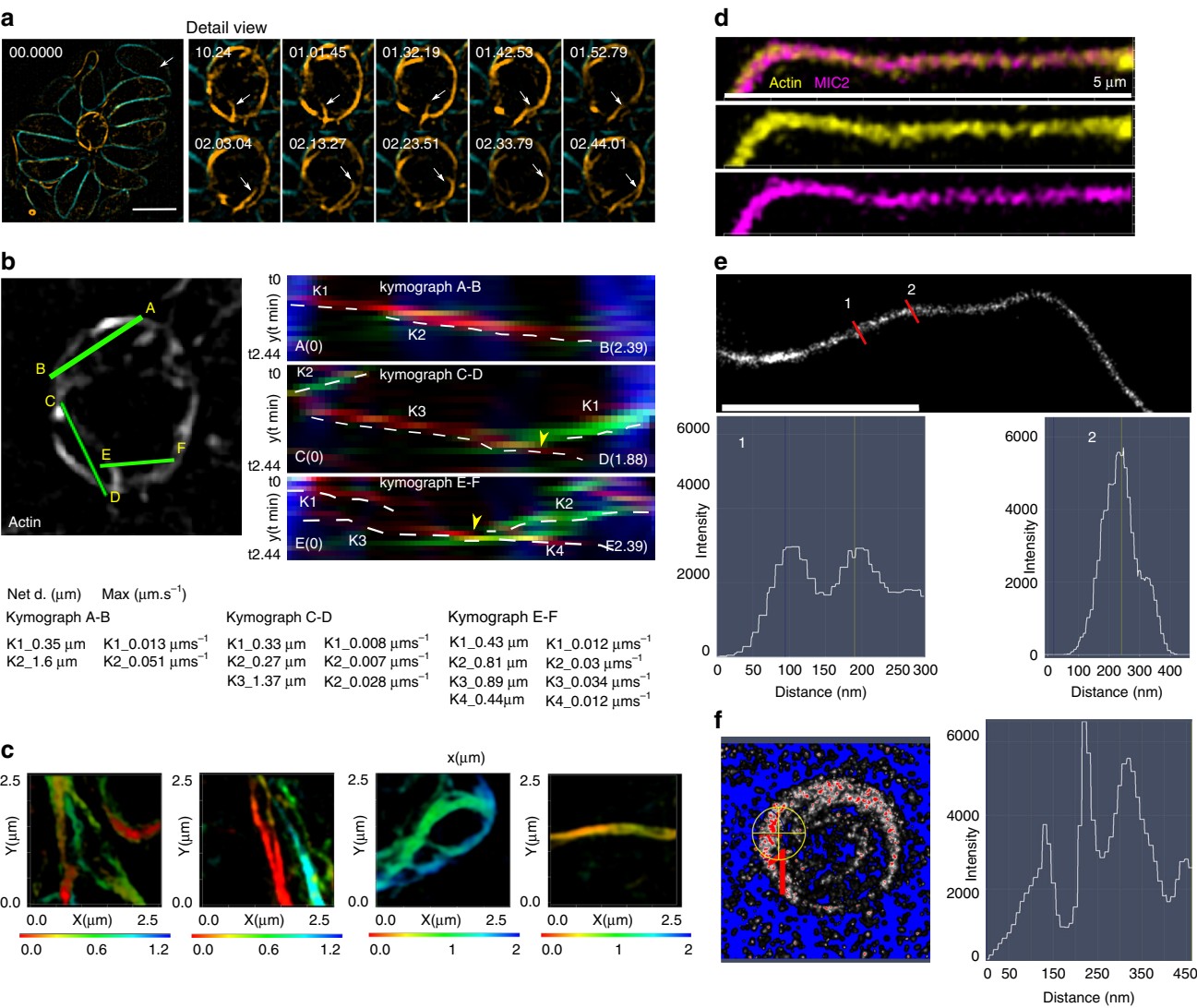

**Fig. 7** Transient self-association and mobility of actin bundles. **a** Representative SIM movie shows transient association and circular flow of actin bundles in the RB of a PV. Detailed view showing SIM frames of the RB. Bundles associate transiently with each other following circular flow in the residual body. The experiment was performed using transgenic parasites expressing actin-Cb-em (orange) and Myo-SNAP visualised with ligand TMR (cyan). **b** Kymograph analyses from the RB (in A) demonstrates that F-actin flow is stationary (blue), anterograde (red) and retrograde (green). Quantitative display of actin flow from measurements in multiple regions of a RB. Visualisation of converging actin flow from multiple parasites supports transient association of F-actin filaments and the formation of the circular scaffold in the RB. **c** 3D-SIM images of the F-actin network. The depth information is colour coded. Data show variable and complex actin bundle association. Representative images are taken from the actin network connecting parasites from three PVs. The experiment was performed on transgenic parasites expressing actin Cb-em. **d** SIM image of an F-actin bundle (yellow) decorated with MIC2-HALO (magenta). A strain stably expressing MIC2-HALO was transfected with actin Cb-SNAP. Ligand for HALO and SNAP were Oregon green and TMR respectively. **e** Representative SMLM image of F-actin bundles connecting parasites in the PV. A PALM experiment was performed using a cell line stably expressing actin-Cb tagged with photoconvertible monomeric mEos3.2 protein (Cb-mEos3.2). This method was a complementary imaging approach to SIM, and was used to resolve the composition of the actin network connecting parasites inside the PV. Variable diameters measured along the actin filamentous structure indicated that the network is an actin bundle consisting of variable number of short F-actin filaments. **f** Representative SMLM image of F-actin bundles forming the RB. A PALM experiment was performed using a cell line stably expressing Cb-mEos3.2. The results show variable diameters of F-actin bundles demonstrating that the RB consists of a network made of bundles with variable number of F-actin filaments. Scale bar is 5 µm. Time in movie frames expressed in minutes. Source data are provided as a Source Data file

(a chloroplast-like organelle)[39] are also directly linked to the secretory system. The apical tip of the parasite also contains the microtubule organising centres from where the subpellicular microtubules are built, which connect to the IMC and are thought to provide the tracks for vesicular transport to the apically localised micronemes and rhoptries[40]. Indeed, disruption of microtubules leads to multiple morphological defects, while actin modulating drugs have less effect on overall parasite morphology[9], which can also be seen in the case of conditional mutants for F-actin[10,13,30] or myosins[11].

These data led to the assumption that the secretory organelles are formed de novo during replication and that microtubule-based vesicular transport is required for the transport of cargo to these organelles[6]. However, with the identification of additional roles of apicomplexan F-actin in apicoplast inheritance[13], dense granule transport[15] and material exchange via an intravacuolar

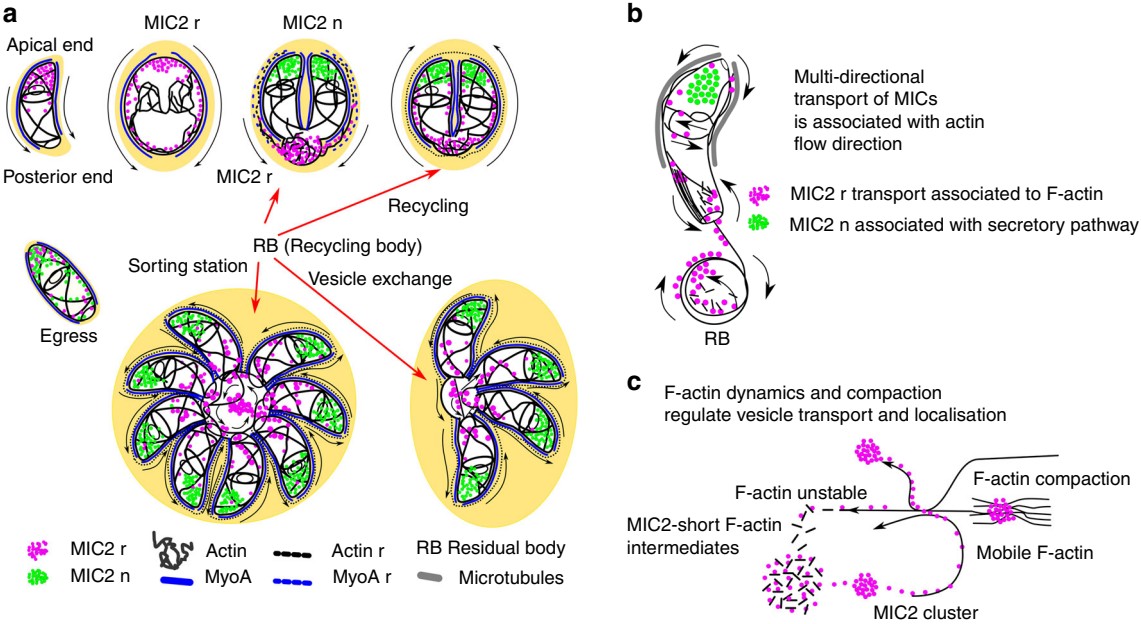

**Fig. 8** Summary of microneme recycling and the role of F-actin. **a** A single cell translocates MICs from the apical to the posterior end during motility and invasion. After invasion the cell formed a PV containing remaining MIC organelles from the mother (MICr). During endodyogeny replication, the cell is disassembled and maternal MIC2 (MIC2r) accumulates at the RB, which acts as a sorting hub for recycled material, while novel MIC2 (MIC2n) is transported towards the apical tip of daughter parasites. At the late stages of replication, recycled MIC2 is transported from the RB to the apical tip of daughter parasites, where MIC2r and MIC2n form two pools of micronemes. With multiple rounds of replication, the RB becomes more and more complex and expanded, connecting individual parasites within a PV. This is a general mechanism used in the replication vacuole for long range, multidirectional transport that is guided by a continuous network of F-actin that connects mother, daughter and the recycling body. **b** F-actin dynamics regulate vesicular transport and cluster formation. We speculate that two populations of MICs can be observed, one formed by recycled vesicles (MICr) that are redistributed using a mechanism depending on F-actin and a second one formed by MICs synthesised de novo and associated with the secretory pathway. Recycled vesicles used F-actin tracks and mobile F-actin for transport and exchange to specific cell locations. **c** Transported vesicles directly interact with F-actin bundles, facilitating long-range transport and cluster formation. The localisation, cluster density and transport kinetics of the vesicles are controlled by the flow, the level of compaction and mobility of the F-actin scaffold

network connecting individual parasites[10,11], we hypothesised that both microtubules and actin are important for vesicular transport during the intracellular development of the parasite. Live imaging of *T.gondii* endodyogeny[5] demonstrated that the maternal secretory organelles disappear around the same time that they are formed within the daughter cells, suggesting that the maternal organelles are either recycled or degraded.

Focusing on micronemes, we examined the mechanisms involved in trafficking of MIC2-positive vesicles during parasite replication. To this end, we designed an experimental approach that enabled us to monitor pools of vesicles synthesised over time, allowing us to discriminate between maternal (=recycled) and novel synthesised MIC2. Furthermore, we identified F-actin as the driving force for material recycling from the mother to the forming daughters and identified the RB as the sorting hub for recycling (Fig. 8a).

In contrast to current models for parasite development, we found that maternal MIC2 is not significantly degraded (or secreted) during parasite replication but is instead almost quantitatively recycled from the mother to daughter parasites. Our data support an even distribution of recycled material in daughter cells within one PV analysed in a time course from 0–18 h.

Tracking vesicle transport in WT parasites suggests that the average speed of recycled vesicles is faster than in those synthesised de novo, which remain more stationary at the apical tip of the parasite. These data strongly suggest an active process of dynamic relocation of maternal material being recycled into newly formed daughter cells. Recycling appears to start early during the replication cycle, with vesicles accumulating at the

posterior end and the nascent RB, before being relocated into the forming daughter cells. We also observed that recycled and de novo vesicles can colocalise and transiently associate with each other, but in all cases recorded, the vesicles never fused, representing two independent subsets. While this indicates that both recycled and de novo vesicles share common receptors for association, transport and clustering with F-actin, they remain independent entities throughout parasite development. This observation might also explain previous studies, suggesting that different subsets of micronemes can be identified in *T.gondii* depending on the trafficking factors, such as Rab-GTPases[41]. It is possible that the two identified subsets are indeed representing recycled vs de novo synthesised micronemes, and that the different, relative content of the vesicles is determined by the timing of expression of the different micronemal proteins (i.e. MIC3,8 appears to be constitutively expressed, while MIC2 expression is cell cycle dependent). In good agreement, the factors identified for differential transport are usually involved in recycling pathways in other eukaryotes, such as Rab5-GTPases[41]. Future studies will be required to further analyse this complex organisation mechanistically.

We also extended this analysis to the IMC, which was previously shown to be recycled from the mother to the daughter at the final stages of endodyogeny[4]. Using an analogous, dual labelling strategy as for MIC2, we found that MyoA, the core motor of the glideosome[42], which is associated with the IMC, is efficiently recycled and appears to follow the same pathway as MIC2. Furthermore, and in perfect agreement with previous studies[31], we demonstrate that nascent MyoA can be found

within the cytosol and later on associates with the IMC of the almost fully assembled daughter cells. Intriguingly, cytosolic MyoA appears to directly associate with the cytosolic side of the IMC. Since it is suggested that the glideosome is anchored between the IMC and the PM of the parasite, these data suggest that at least during this replication stage the IMC is still permeable and allows passing of the glideosome to the outer leaflet. In good agreement, recycled IMC material containing MyoA appears to be transported to the RB using cytosolic F-actin as tracks, again suggesting that the glideosome has access to the cytosol of the parasite and might lose its anchorage to the PM. While we demonstrate that the recycling pathway depends on F-actin dynamics (see below), we would like to point out that MyoA is probably not the motor protein driving its own recycling, since *myoA* mutant parasites do not show any defects in parasite development[13,43] that are comparable to *act1* cKO. Instead, we speculate that MyoF, the only conserved myosin in apicomplexans apart from MyoA[44], is required for multiple recycling pathways, since it has been demonstrated that this myosin is required for dense granule trafficking and apicoplast inheritance and, in good agreement with a role in recycling, accumulates in the RB when expressed as dominant negative[15,16].

Using synergistic super-resolution and live imaging approaches, we provide evidence for the crucial role of F-actin in vesicular transport and recycling of maternal material: (a) SIM imaging demonstrated that MIC2-positive vesicles perfectly aligned and localised to F-actin bundles; (b) co-localisation and cluster analysis based on SMLM, a method that extracts localisation position of molecules with a precision ranging 20–50 nm[45] fully validated association of MIC2-clusters to F-actin and that treatment with actin-modulating drugs (Jas or CD) changes cluster size. (c) kymograph and kinetic analysis of both F-actin dynamics and vesicle movement demonstrate flow of actin and MIC2 vesicles, that they co-localise and share directionality and kinetics. Furthermore, convergence of F-actin flow results in MIC2 cluster accumulation at the intersection; (d) using triple labelling and SIM microscopy, we demonstrate that F-actin and the subpellicular microtubule form a continuous network underneath the IMC, with actin concentrated at the posterior pole. The observations are limited by the resolution of 3D SIM microscopy (100–130 nm in the lateral and axial plane and 250–350 nm in the axial plane)[45]. Importantly, MIC2-positive vesicles were identified that interact either with actin, microtubules or both, giving an explanation for the presence of F-actin independent MIC2-clusters. Furthermore, it suggests a cooperation and coordination of F-actin and microtubule-based vesicular transport during daughter cell development. These findings agree with recent reports that establish the complex regulation between actin and tubulin, including physical interaction between the plus end of microtubules that results in actin polymerisation, the block in microtubule polymerisation in the presence of actin meshwork[46,47], and transport of actin on a microtubule network[48].

In summary, our observations support a role of F-actin in two types of behaviour (Fig. 8b): (a) as physical tracks for long range, multidirectional transport within the cell and the extracellular parasite network connecting the daughter cells with the RB playing a major role in actively exchanging material; (b) as highly dynamic mobile bundles that facilitate exchange of material and interconnect distant cells inside the PV in a way not being dissimilar, phenotypically and structurally, from cytonemes - filopodia like structures found in higher eukaryotes that transport proteins and are involved in cell communication[49,50].

In summary, our data support a model, where a highly dynamic F-actin network that connects daughter cells within a PV acts as the driving force for recycling of maternal organelles,

such as micronemes or the IMC. In this model the RB is the sorting hub for recycling and material exchange between cells (Fig. 8). SMLM measurement furthermore suggests that the impressive F-actin dynamics is organised by the rapid association and disassociation of (probably short) F-actin bundles that are associated with transport vesicles, which are directly transported within the dynamic F-actin network. Therefore, the actin connectivity between mother and daughters provides a structure that ensures that the parasite uses its resources highly economically by recycling maternal material. We show a general mechanism by a non-canonical actin that regulates vesicle transport and localisation of vesicles by changes in compaction and self-association of its highly dynamic scaffold. Our data also suggest a highly coordinated interplay between the subpellicular microtubules and F-actin in vesicular transport in apicomplexan parasite.

## Methods

**Culturing of parasites and host cells**. Human foreskin fibroblasts (HFFs) (RRID: CVCL_3285, ATCC) were grown on tissue culture-treated plastics and maintained in Dulbecco's modified Eagle's medium (DMEM) supplemented with 10% foetal bovine serum, 2 mM L-glutamine and 25 mg/mL gentamycin. Parasites were cultured on HFFs and maintained at 37 °C and 5% CO2. Cultured cells and parasites were regularly screened against mycoplasma contamination using the LookOut Mycoplasma detection kit (Sigma) and cured with Mycoplasma Removal Agent (Bio-Rad) if necessary.

**Plasmid construction**. To endogenously tag *MIC2*, the 3′flank of the gene was amplified using primers FW 5′ TACTTCCAATCCAATTTAATGCAGTGCCGA AGCATGTGTGGGTGCA3′ and RV 5′CCTCCACTTCCAATTTTAGCCTCCA TCCACATATCACTATCGTCATCCACGGG3′ and inserted by ligation independent cloning into a LIC vector[21] with the GFP tagged replaced with HALO using primers FWPG1-HALO 5′ATTACCTAGGAGGATCTGTACTTTCAGAG3′ RV-PG1-HALO 5′ATTACTGCAGTT AACCGGAAATCTCCAGAGTAG3′ generated from a Cb-HALO vector template[10].

To create Cb-SNAP, the open reading frame from vector template P5RT70-Myc-GFP[51] was replaced and subcloned with Cb and SNAP tag encoding sequences using standard restriction digestion/ligation protocols. The CB sequence was amplified using primers (FW-Cb 5′ATTA GAATTCCCTTTTTCGACAA AATGGCTCAGGTGCAGCTGGT3′ RV-Cb 5′TAATCT GCAGCGCTTCTTGA GGAGACGGTGA3′ using Cb-HALO as a vector template[10], followed downstream by a SNAP coding sequence in frame, that was amplified using primers FWPG1-SNAP: 5′ATTACTGCAGGAATGGACAAAGACTGC GAAATGAAGC3′ and RVPG1-SNAP: 5′TATGTTAATTAATCAACCCAGCCCAGG CTTGCCCAG3′ from a SNAP plasmid (NEB)[29].

To create Cb-mEos3.2, the sequence encoding HALO from vector template Cb HALO[10] was replaced with a sequence encoding mEos3.2[28] from a plasmid template mEos3.2-lifeact-7, a gift from Michael Davidson (addgene plasmid 54696; http://n2t.net/addgene:54696; RRID: Addgene_54696. amplified using FwmEos3.2 ATTACTCGAGGAGGGGGGATCCACCGGTCGCCACC, RvmEos3.2 5′TAT GTTAATTAATCATTATCGTCTGGCATTGTCAGGC

To create Myo-SNAP vector, PMyoA-Ty-MyoA[14] was used as a template backbone and the Ty sequence replaced in frame with a sequence encoding SNAP from plasmid template SNAP (NEB) amplified with primers FWSNAP 5′ TAATGAATTCCGACAAAATGGACAAAGACTGCGAAATGA3′, RVSNAP 5′ TAATATGCATACCCAGCCCAGGCTTGCCCAG3′.

***T. gondii* transfection and selection**. To generate stable actin Cb-mEos3.2, $1 \times 10^7$ of freshly released RH Δ*hxgprt* parasites were transfected with 20 µg plasmid DNA by AMAXA electroporation.

To generate Cb-Em/MyoA-SNAP, actin Cb-Em expressing parasites, the same method was followed using as a parental strain[10]. MIC2-HALO and MIC2-HALO/ Cb-em strains were generated using RH Δ*Qu80Δhxgprt*. Transfected parasites were then sorted using flow cytometry with a S3 Cell Sorter (Bio-Rad, Hercules, CA, USA). A schematic of the constructs and strains are shown in Supplementary Fig. 6

**Fixation**. Samples were fixed using a reconstituted solution consisting of two different buffers: Cytoskeleton buffer (CB1) (MES pH 6.1 10 mM, KCL 138 mM, MgCl 3 mM, EGTA 2 mM, 5% PFA) and Cytoskeleton buffer (CB2) (MES pH6.1 10 mM, KCL 163.53 mM, MgCl 3.555 mM, EGTA 2.37 mM, Sucrose 292 mM). Buffers were mixed in a 4:1 ratio, respectively, and used for fixation for 25 min. After that, samples were treated with a PFA quenching solution (NH4CL 50 mM) for 10 min, and washed three times with PBS.

**Recycling of MIC2-HALO in replicating PVs**. Recycling of MIC2-HALO pools was done using three types of independent but complementary imaging

experiments. Time course dual colour labelling (dyes and antibodies used in this study summarised in supplementary table 1) in SIM fixed assays, dual colour labelling imaged with diffraction limited wide field, live microscopy, and time course experiments of single colour SMLM.

MIC2-HALO recycling in PVs was investigated using dual colour labelling SIM. Host cell monolayers infected with 48 h PV expressing MIC2-HALO were labelled with HALO ligand coupled with TMR at a concentration 160 nM for 30 min at 37 °C. Then, the dish media were changed three times and left in fresh media for at least 2 h. A final change of media was added before imaging to remove dye traces in the solution. Parasites were then scratched, syringed, centrifuged at 600 G for 5 min, and resuspended in fresh media to infect, live imaging dishes. Independent dishes were labelled after intervals of 0, 6 and 18 h with a second dye, Silicon-Rodhamine, at the same concentration (STFC) following the same labelling procedure, washed for at least 2 h, and fixed. At least five fields of view (FOVs) were recorded per time point.

Reproducibility of the recycling results from SIM imaging was ensured by performing two additional independent experiments. First, using only one single colour labelling experiment (TMR ligand), PVs were imaged at an 18 h time point, and a dual colour labelling experiment using as a second dye Oregon green (at similar concentration 160 nM) added after 24 h. At least five FOVs were recorded per time point. The data were imaged with SIM.

MIC2-HALO recycling kinetics were monitored with live imaging using diffraction limited wide field fluorescence at 37 °C. Triplicate PV parasites expressing MIC2-HALO were labelled following the same procedure but using pair dyes TMR/Oregon green (Promega) in 14 h interval.

Time course of single colour labelling of MIC2-Halo was labelled with TMR in PV at 0, 6, 12 and 24 h. triplicate PVs were analysed for each time point, fixed and imaged with SMLM. A t-test, and a one way ANOVA, Tukey's test comparing vesicle numbers were performed and analysed using GraphPad Prism7.0.

**Chase experiment of MIC2-HALO.** Twelve hours PVs expressing MIC2-HALO were labelled sequentially with TMR, and Silicon-Rodhamine, respectively, in a 12 h interval, after which cells were fixed. At least six representative field of views were recorded with SIM. The experiment was repeated in similar conditions to perform live SIM imaging experiments. At least four movies with PVs in different cell stages were recorded. Additionally dual labelling experiments were performed in a parasite lines expressing MIC2-HALO and actin Cb-em. At least 12 FOVs representative of different PV stages were imaged with SIM.

**Vesicular transport and association of MIC2-HALO vesicles on actin filaments.** PVs in different replication stages (1, 4, 8, 32 cells) expressing MIC2-HALO and Cb-em were labelled with TMR dye as described above and imaged with live SIM imaging in at least three independent experiments (22 movies), and diffraction limited wide field fluorescent microscopy for one experiment (four movies). Estimated number of vesicles and ratios of de novo (n) and recycled (r) in SIM images were calculated at incremental distance to the actin bundle based on a voxel size with dimensions 0.0322 µm (x, y dimension of a voxel), 1.1 µm (the z dimension of a voxel). Measurements were taken at 0.55 um (5 voxels), 1.1 um (10 voxels), 2.2 µm (20 voxels) and 3.3 µm (30 voxels). Images were first adjusted to have a gamma of 1.0 followed with an adjustment such that the 90% of the pixels with highest intensity were retained and the pixels which were in the lowest 10% of intensity values were removed (background). Threshold for rendering was then adjusted manually to ensure that every volume above background intensity was recognised by placing a rendered spot at the centre of the volume, or rendering a surface on the actin volume. Actin bundle is defined here as surfaces that include only large intact actin chromobody structures (>1, 5 µm²) and is called actin chromobody bundle.

To test the effect of actin depolymerisation on MIC2 localisation. 16–32 cells PVs were treated with 1 µM CD at 37 °C for 1 h or left untreated (control), fixed and imaged with 2D-SIM. 11 well defined PVs were analysed per condition from 9 independent FOVs, and three intensity profiles taken per PV from the apical end to the posterior end. To investigate the localisation of MIC2 in single parasites, parasite line Cb-em-MIC2-HALO was also stained with a live marker for microtubules (Sir-Tubulin) following manufacturer's instructions and imaged in live (one experiment) and fixed SIM (at least three independent experiments).

**Localisation and recycling of MyoA-SNAP in replicating PVs.** Twenty-four hours PV expressing MyoA-SNAP and Cb-em were labelled with TMR dye at a 120 nM concentration, washed overnight and imaged after 36 h. The experiment was repeated with SIM live imaging in three independent experiments and at least five movies were taken in each experiment. The experiment was repeated with sequential labelling of Myo-A using Alexa-647 at the same concentration in two additional independent experiments.

Recycling of MyoA-SNAP in replicating PVs was monitored independently by labelling 48 h PVs expressing Myo-SNAP with 505* dye, left washing and to replicate overnight, fixed and co-stained with inner membrane complex marker antibody IMC1 1/1000 dilution as described in Periz et al.[10].

Intensity profiles were calculated as a percentage of the maximum intensity signal for each channel, and represents the relative spatial distribution of the fluorescence intensity associated with a specific labelled protein.

**Plaque assay.** Conducted as described in Periz et al.[10]. In all, $1 \times 10^3$ parasites were inoculated on a confluent layer of HFF and incubated for 5 days, after which the HFF were washed once with PBS and fixed with ice cold MeOH for 20 min. HFFs were stained with Giemsa with the plaque area measured using ImageJ. Mean values of three independent experiments ± SEM were determined. A minimum of 15 FOVs were analysed. The surface was expressed in pixel units.

**Trail assay.** Gliding assays were performed as described[10]. Briefly, freshly released parasites were allowed to glide on FBS-coated glass slides for 30 min before they were fixed with 4% PFA and stained with α-SAG1 under non-permeablising conditions. Mean values of three independent experiments ± SEM were determined.

**Live cell 2D motility.** Both RH-Cb-emerald (control) and MyoA-SNAP/Cb-emerald parasites were artificially released using a 23 G needle and filtered, spun down and resuspended in pre-warmed gliding buffer (1 mM EDTA, 10 mM HEPES in HBSS). These were then added onto FBS-coated glass bottom dishes and transferred to the DV core microscope (Applied Precision, GE). The cells were maintained under standard culturing conditions and imaged at two images per second using a ×40 objective lens. Image sequences were analysed using the Icy Image Processing software (Institut Pasteur) with the Active Countours plugin[52]. Average distance and speed were calculated for 10 parasites exhibiting helical or circular motions. Statistics were analysed using a non-parametric Mann-Whitney test in GraphPad Prism7.0.

**Wide field fluorescent live imaging.** Live imaging was recorded with a Delta-Vision Core microscope (Applied Precision, GE) attached to a CoolSNAP HQ2 CCD camera. An Olympus UPLSAPO 100 × oil (1.40NA) objective, maintained throughout the experiment (37 °C; 5% $CO_2$). Images were deconvolved using SoftWoRx Suite 2.0 (Applied Precision, GE). Image processing was performed using FIJI for ImageJ (NIH), Icy Image Processing Software (Institut Pasteur).

**SIM imaging.** SIM in fixed assays was taken using Zeiss Elyra PS.1 system as described[10]. For Live SIM imaging, data were taken with an objective C-Apochromat Korr M27 at room temperature and performed using a software module adapted for a multi-purpose filter set dual for PALM/dSTORM and SIM (Zeiss). The filter set consisted of adjusted filter cubes with the following major beam splitters, laser-blocking and emission filters: position 5: MBS 405/488/561/642 + LBF -561/642 (used with laser lines 561, 642); position 6: MBS 405/488/561 + LBF -488/561 (used with laser lines 488, 561). 3D rendering and models views were generated using Zen software and Imaris software (Bitplane, Oxford Instruments).

**SMLM cluster analysis.** Processed reconstructed dSTORM images were analysed using MATLAB SMLM cluster and colocalisation software called Clus-DoC (Cluster detection with Degree of Colocalisation)[26]. Vacuoles containing 2–8 cells were selected and a number of tests performed to first detect and quantify clusters then calculate the degree of colocaliation between them. A Ripley K test was performed to obtain a maximum L(r)-r value for each vacuole. This value was input into the DBSCAN and DoC test parameters to filter out noise points. To obtain the normalised molecular density maps of MIC2, a DBSCAN test was run on individual cells from each treatment group with an *Epsilon* value of 10 nm and *MinPts* value of 3. The relative cluster density is a measure of the local variance of molecular density within the defined clusters To quantify the degree of colocalisation between MIC2 and Actin a DoC test was run on entire vacuoles from each treatment group. For each molecule of protein A, the number of molecules of protein A and B within circles of increasing radius were calculated, giving the density gradients of protein A and B around the original molecule. The two distributions were compared by Spearman's rank correlation and each molecule assigned a DoC score. These scores range from −1 (anti-colocalised), through 0 (no colocalisation), to +1 (perfectly colocalised). These results were combined with DBSCAN results to segregate clusters into colocalised (>3 molecules with DoC greater than 0.4) and non-colocalised (<3 molecules with DoC greater than 0.4) groups. Properties such as cluster size and colocalisation percentages could then be extracted and DoC frequency histogram plots produced.

**Dual colour SMLM (dSTORM).** For dual colour SMLM, a MIC2-HALO stable cell line was transfected transiently with actin Cb-SNAP, left replicating overnight and stained with TMR-HALO and Alexa 647-SNAP respectively, after which the samples were incubated with either 100 nM jasplakinolide or 2 µM CD D for 1 hr at 37 °C or left untreated (control) and fixed. PVs containing 2–8 cells (at least three vacuoles per each treatment) were imaged with SMLM. Two colour, two-dimensional (2D) dSTORM imaging was carried out using a super-resolution Zeiss Elyra PS.1 system equipped with a 1.46 NA ×100 oil immersion objective (ZEISS

Alpha Plan-Apochromat ×100/1.46 Oil DIC). dSTORM imaging of TMR and Alexa Fluor 647 was accomplished with 561 nm and 642 nm wide-field laser excitation, respectively. The residual laser excitation beams in the imaging path were filtered out using a multi-band laser-blocking filter (561/642 nm). HILO-fluorescence illumination was used to minimize axial excitation and reduce background, thereby enhancing signal to noise ratio (SNR) in the images. Prior to dSTORM data acquisition a single frame snapshot of wide-field image was taken for each excitation wavelength for comparison purpose. A maximum of 40,000 frames were collected using the exposure times of 20 ms. The EM gain in the EMCCD camera (Andor iXon 897) was set at 200–300% depending on the SNR. The drifts in axial direction were corrected in real time using definite focus function. A photo-switching buffer of 100 mM mercaptoethylamine in PBS was mounted on top of the samples during data acquisition.

ZEISS ZEN software was used to process and render dSTORM images. In ZEN, point spread function (PSF) mask size (typically 9 pixels) and intensity to noise ratio (typically 6) were defined firstly, and then each blinking event was localized using a 2D Gaussian fit model. Overlapping fluorophores were accounted for using the Account for overlap setting. Following the localisation, the displacements of molecules in the lateral plane from drifts in the reconstructed images were corrected for using feature detection and cross correlation methods. Channel alignment was implemented using a standard multi-colour TetraSpeck bead sample. Therefore the aberrations in the reconstructed images from misalignment between the two colour channels were reduced accordingly.

**Particle tracking based on kymograph analysis**. Trafficking movies were ana-lysed using the Icy platform for Bioimage analysis[53]. Movies were selected based on clarity of particle displacement. Region of interest (ROI) tracks were drawn in places of particle movement and then analysed by the KymographTracker plugin for Icy[52]. For the analysis, 5 pixels were selected for the width of the analysed tracks. The kymogram was generated without separating particle direction and then analysed by drawing ROI tracks on the patterns of moving particles. The information was then analysed by the Icy TrackManager plugin "MotionProfiler", which can read the drawn ROI and extract information regarding speed, intensity and displacement. This process was done for both micronemes and actin particles using the same drawn tracks for comparison. The extracted values were then exported to Microsoft Excel for further analysis. A more detailed manual can be found on the plugin's website: http://icy.bioimageanalysis.org/plugin/KymographTracker.

**Color-coded kymogram generation for particle dynamics analysis**. Color-coded kymograms were generated using the KymographClear plugin on ImageJ[54]. This plugin allows to trace a path of interest on a collapse t-stack to follow particle movement. The plugin is able to generate a three color-coded kymogram using a Fourier algorithm to differentiate populations of particles moving in distinct ways. The color-coding is represented by forward movement as red, backward movement as green and no movement as blue. A more detailed protocol can be found on the author's website: https://sites.google.com/site/kymographanalysis/.

**Reporting summary**. Further information on research design is available in the Nature Research Reporting Summary linked to this article.

## Data availability

All data and genetic material used for this paper are available from the authors on reasonable request. The source data underlying Figs. 1d, e, h, 2, 3f, g, 4a, d, e, 5d, h, 6d, o, 7b and supplementary Figs. 2, 3, 4b, 5d are provided as a Source Data file.

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

## Acknowledgements

J.P. was supported with a Central Laser Facility Octopus facility access award (STFC, Rutherford Appleton laboratory). MM is funded by a Wellcome Trust Senior Fellowship (087582/Z/08/Z) and an European Research Council starting Grant (ERC-2012-StG 309255-EndoTox. The Wellcome Centre for Integrative Parasitology is supported by core funding from the Wellcome (085349). Dr Alana Hamilton with help to purify strains with Flow cytometry facility at WCIP. Dr Leandro Lemgruber from WCIP Imaging facility for maintenance of microscope set ups. Professor Gary Ward provide antibodies used in this work. We thank Mrs Frances Maxwell for MATLAB and editorial assistance.

## Author contributions

J.P. designed, performed and coordinated the experiments, analysed the data, created all schematics in the figures and wrote the paper; M.D.R. contributed reagents, analysed the data; A.M.S. performed SML experiments and analysed the data; S.G. contributed reagents; C.L. analysed SIM data; L.W. performed the SML experiments analysed the data, writing—review and editing; M.L.M-F. provide the resources, writing—review and editing, coordinate the project; M.M. designed and coordinated the project and experiments, analysed the data, contributed resources and wrote the paper.

## Additional information

**Competing interests:** The authors declare no competing interests.

