## [Peer Review File · Nature Communications]

Reviewers' Comments:

Reviewer #1:

Remarks to the Author:

In the manuscript entitled "A highly dynamic F-actin network regulates transport and recycling of micronemes in *Toxoplasma gondii* vacuoles", J. Periz and colleagues use advanced light microscopy techniques to document the process of recycling micronemes in *T. gondii* parasites as well as the role of actin cytoskeleton in this process. The authors conclude that the micronemes of the mother cell are recycled into the daughter parasites and that this is possible thanks to an extensive F-actin network connecting daughter parasites. While the study is novel and provides important insights into the parasite biology, the authors ought to provide some extra evidence to support their claims.

Techniques: The authors use state-of-the-art light microscopy techniques and extensively describe the methods they used. The microscopy data presented in this manuscript are of impeccable quality and the authors present appropriate quantifications. In addition, the authors use an elegant dual-labelling approach to study the fate of new and recycled vesicles. However, could the authors please explain the statistical tests used in Fig. 1d,e,i and Fig. 3k,l?

Conclusions and novelty: The study is novel and the conclusions with regard to microneme recycling are supported by a good body of evidence. These studies take advantage of an elegant use of a dual-labelling approach. The authors also show a great amount of data in support of the role of actin cytoskeleton in microneme recycling and vesicle trafficking in the parasite. While the authors use state-of-the-art light microscopy to demonstrate the association of MIC2 vesicles with actin cytoskeleton, they are missing a couple of important controls required to support their claims. In particular, the following experiments are missing:

1. The authors claim that the MIC2 recycling and trafficking are microtubule-independent. However, they do not include a simple experiment using microtubule modulators in support of their claims, despite describing in the introduction that it is believed that the daughter cell assembly is facilitated by microtubules. This is particularly important in the context of actin association with subpellicular microtubules they show in the later part of their manuscript. The authors should test whether microneme recycling as well as MIC2 localisation is affected by microtubule-modulating drugs.
2. The authors use actin modulators to affect the dynamics of actin cytoskeleton. This leads to reduction in MIC2 cluster density, especially upon depolymerisation of actin filaments (Fig. 3i). However, despite the seemingly reduced amounts of MIC2, the distribution remains similar to the control (especially at the apical end, Fig 3h). Could the authors please quantify the changes in MIC2 levels and its distribution and comment on the fact that despite actin depolymerisation/stabilisation, MIC2 vesicles still reach the apical end? One could speculate that the observed apical signal represents MIC2 that reached the apical end before the cytochalasin D/jasplakinolide treatment. This could be addressed if the authors use the same dual labelling approach they utilised before: they could label MIC2 with one colour, perform the treatment with actin modulators and then label the newly synthesised MIC2 with another colour to look specifically at the population synthesised/trafficked after the treatment with actin modulators.
3. The authors should at least discuss or ideally perform a simple experiment to test whether the actin-dependent mechanism for microneme recycling is specific or whether it reflects a global problem with protein trafficking in the cell in the absence of actin filaments (either because of lack of actin highways for the vesicles or even failure in vesicle formation). The authors could test this by checking what happens to the localisation of several reporter proteins in parasites treated with actin

modulators, fixed and stained with specific antibodies.

4. The authors describe differential distribution of the new vs recycled vesicles (Figure 2) and state: "An estimation of the total number of MIC2r and MIC2n indicated that MIC2 vesicles represent two distinct sub-populations: MIC2n can be found mainly at the apical tip of the parasite, while a significant amount of MIC2r appears to be associated with actin bundles localised at the posterior pole (Fig. 2 d,e)." This should be quantified.

I believe that once the authors provide these additional evidence in support of their hypothesis, the manuscript will be suitable for publishing in Nature Communications and provide a very valuable resource for the scientific community. In particular, the imaging techniques used in this work will surely inspire other groups and will provide a fantastic guide on how to perform cutting-edge microscopy on intracellular parasites.

Reviewer #2:

Remarks to the Author:

In this manuscript, the authors show the recycling of microneme vesicles from the mother to the daughter cell during *Toxoplasma gondii* replication. Using cutting-edge microscopy, they were able to show that the recycling of vesicles is dependent of actin located inside the parasite as well as in the residual body. The finding is novel and achieved using a technically solid approach, but the overall manuscript needs to be clearer to be understandable for the reader and some points need to be addressed.

Major comments

- 1- What is the ratio of de novo vesicles associated with actin compared to recycled? Are the de novo synthesized vesicles using actin or are they directly made at the apical of the daughter cell?
- 2- How frequent are the 2 types of vesicular transport dependent of F-actin? Especially the one where the actin self-associate to nearby bundles and form bridge (figure 3e)?
- 3- The authors state that actin dynamics is required for IMC recycling, but they show it only for MIC using CD and JAS. Is there any reason why the authors did not check the association of MyoA and actin in presence of CD and JAS?
- 4- Do the authors have evidence that the actin connects the apicoplast and the trans-golgi network through their data or through literature? In the latter case, a citation should be added and it should not be stated in the legend figure 5e,f,g without data support.
- 5- The authors can't conclude based on their experiment that there is an interplay between the subpellicular microtubules and F-actin in vesicular transport. The authors only show that they are in close proximity and also said in the text that because there is no microtubule in posterior areas of the parasite, it suggests a microtubule independent transport mechanism. They also can't conclude in the legend of figure 5l,m,n that the microtubule connects with actin. Furthermore, what is the conclusion of the long microtubule polymers observed in figure 5h for vesicle transport?

Minor comments

- 1- Figure 1b – what are the numbers in the picture corresponding to?
- 2- The author needs to change this sentence to make it clearer. "We also confirmed that micronemes are formed de novo, since after each round of replication the ratio of recycled versus de novo synthesised MIC2-Halo reflects an even distribution of recycled and de novo synthesised material in individual parasites within a PV (Fig.1b)."
- 3- The authors should state in the text that the figures 1c,d,e are on recycling vesicles only as their previous experiments are on both recycled and de novo vesicles.

- 4- This sentence is not clear, what does the author wanted to say? Please, reformulate the sentence to make it clearer. "Only a slight non-significant variation occurs after 24h suggesting a level of heterogeneity of maternal microneme vesicles due to processing or different degree of cluster formation (Fig.1e)."
- 5- Figure 1h and 3, the unit for the video time should be mention somewhere, for example in the legend.
- 6- Figure 1i, the unit of the transport speed and trajectories should be added in the text when mention. "Pools of recycling vesicles have slightly faster transport (0.267 ± 0.003 versus 0.232 ± 0.004) and larger trajectories (0.421 ± 0.143 versus 0.375 ± 0.16) than de novo synthesised vesicles (Fig. 1i)."
- 7- Figure 1i (actin not labelled) and figure 3f (chromobody), the speed number is considered similar (0.267 and 0.37 to $1.27\mu\text{s}^{-1}$) while the author considered that 0.267 to $0.232 \mu\text{s}^{-1}$ is different. The same comment applies for the trajectories number. Could the author clarify their statement, maybe with statistical analysis?
- 8- For all the figures, especially for figure 3a, 3i and figure 5, it will be useful to have the apical/basal orientation of the parasite as the pictures are not in the same orientation (keeping the same orientation inside a figure will help the reader).
- 9- The word 'with' need to be deleted. "In summary, we obtained unambiguous data from fixed images, showing examples of small and big parasitophorous vacuoles, with using live SIM and conventional wide field imaging (Fig.3), we confirmed that MIC2-containing vesicles are associated to a very dynamic F-actin network that connects the posterior end of parasites with the residual body (RB; (Fig. 3a, b, c, Supplementary Fig. 1, Supplementary Video 3-5)."
- 6- Figure 3 the mic2 vesicles labelled are only recycled or recycled and de novo?
- 7- Figure 3i, the author should be consistent with the use of abbreviation (CD rather than cyt-D) as well as in the M&M in dual colour SMLM where they used CD D.
- 8- Figure 3h the authors said "Right. Intensity profiles of PVs treated with CD show loss of actin network, concentration and re-localisation of MIC2 vesicles to the apical end periphery." The figure shows the concentration and re-localisation of MIC2 vesicles but not the loss of actin, the authors should reformulate the sentence to say that PV treated with CD where actin network has been loss, show a concentration and re-localisation of MIC2 vesicles.
- 9- The author never cited the figure 4h. Furthermore, the colors used in the intensity profiles are difficult to distinguish and there is no mention of their meaning.
- 10- Figure 5d and o, the orientation of the profile should be added as it has been done in other figures (ie from a to b).
- 11- Figure 6a, the color code is not mention and the time for the video need to be written somewhere.
- 12- The schematic model should be better explained in the legend, especially that the unstable actin (due to CD for example) induce clustering of vesicle. The two different actin dependent transport are not clearly describeds.
- 13- In the M&M, the author wrote Q80 instead of ku80 (T.gondii transfection and selection section).
- 14- For the recycling of MIC2-HALO in replicating PVs, the authors mention that they washed for at least 2h more, what does it mean? They also talk about FOVs but never explain the abbreviation in the manuscript, that likely stand for field of view.
- 15- Supplementary figure1, the legend a is for a movie that is not there, inducing a shift in the following letter.
- 16- The author never mentions the time of replication of Toxoplasma (6h), that could help understanding the time point they used in different experiments for reader outside of the field.

Reviewer #3:

Remarks to the Author:

This work presents stunning images and the conclusion of the results on the re-cycling of micronemes and likely other organelles via an F-actin network through the residual body is novel and important for the apicomplexan biology field.

The techniques used based on snaptags, halotags, allowing chasing of micronemes is also novel for Toxoplasma and the images and videos are of amazing quality.

Critique:

It will be a lot clearer if all images could be separated. For example some of the expanded views of fig 2, b, c, 3b, g, d, e, 4f, 5c, 6b, and others that may be overlapped. Sometimes it is hard to distinguish single from two overlapping images.

Several figures refer to representative images and one wonders about the number of experiments performed so it will be good to include for each figure the number of experiments performed (n).

Please provide statistical analysis when appropriate.

Is MIC2-Halo secreted normally? Is trafficking altered by the tagging?

It will be good to include controls for the MyoA-Snap: growth, motility, etc. The cells in Figure 4b look a little swollen.

What is the meaning of F-actin bundles encasing the MICr clusters? Wouldn't the clusters move along the network?

Panel D from fig S1 is missing

In Fig 3 why does Cytochalasin D and JAS both increase the area considering that they have opposite effects? An explanation would be good.

Figure 4 is confusing about which ones were done with live or fixed cells. Could this be clarified?

Does CD and Jas affect MyoA traffic also?

A sporozoite is mentioned in Figure 5, should it be tachyzoite?

Could the authors provide cartoons showing the strategies for tagging in the supplemental material?

Why there is not a separate discussion?

Point-to-Point Reply to the reviewers

We would like to thank the reviewers for their efforts and valuable comments that have significantly contributed to improve the quality of our study.

Please find our reply in blue:

Reviewer #1 (Remarks to the Author):

In the manuscript entitled "A highly dynamic F-actin network regulates transport and recycling of micronemes in *Toxoplasma gondii* vacuoles", J. Periz and colleagues use advanced light microscopy techniques to document the process of recycling micronemes in *T. gondii* parasites as well as the role of actin cytoskeleton in this process. The authors conclude that the micronemes of the mother cell are recycled into the daughter parasites and that this is possible thanks to an extensive F-actin network connecting daughter parasites. While the study is novel and provides important insights into the parasite biology, the authors ought to provide some extra evidence to support their claims.

Techniques: The authors use state-of-the-art light microscopy techniques and extensively describe the methods they used. The microscopy data presented in this manuscript are of impeccable quality and the authors present appropriate quantifications. In addition, the authors use an elegant dual-labelling approach to study the fate of new and recycled vesicles. However, could the authors please explain the statistical tests used in Fig. 1d,e,i and Fig. 3k,l?

Conclusions and novelty: The study is novel and the conclusions with regard to microneme recycling are supported by a good body of evidence. These studies take advantage of an elegant use of a dual-labelling approach. The authors also show a great amount of data in support of the role of actin cytoskeleton in microneme recycling and vesicle trafficking in the parasite. While the authors use state-of-the-art light microscopy to demonstrate the association of MIC2 vesicles with actin cytoskeleton, they are missing a couple of important controls required to support their claims. In particular, the following experiments are missing:

1. The authors claim that the MIC2 recycling and trafficking are microtubule-independent. However, they do not include a simple experiment using microtubule modulators in support of their claims, despite describing in the introduction that it is believed that the daughter cell assembly is facilitated by microtubules. This is particularly important in the context of actin association with subpellicular microtubules they show in the later part of their manuscript. The authors should test whether microneme recycling as well as MIC2 localisation is affected by microtubule-modulating drugs.

This is a very good point and we thank the reviewer for this suggestion. Based on our data we can conclude that the recycling (but not *de novo* biogenesis) of micronemes is actin dependent. We are currently performing experiments to analyse the role of microtubule dependent transport during parasite development. Unlike the dynamic, labile, and idiosyncratically arranged microtubule cytoskeleton in mammalian cells, *T. gondii* cortical microtubules are exceptionally stable and have the same distribution in every cell. They do not display dynamic instability and are not even depolymerised when the free tubulin concentration is drastically reduced by detergent extraction ¹;

2. Indeed, most drugs affecting microtubules are not working on apicomplexan microtubules. The only drug used in the field is Oryzalin. However, it can only block formation of the subpellicular microtubules and therefore leads to disruption of daughter cell development. Parasites then appear pleiomorph with multiple defects. Furthermore, microtubules of the host cell are also affected complicating matters for imaging. To circumvent this problem, we generated conditional mutants for alpha-Tubulin using cas9 (see images below for the reviewer's information) for an independent study. Unfortunately, the effects are also pleiomorph and cannot be interpreted in any reliable manner, especially with regards to vesicular transport.

Most importantly, there are no individual parasites, but instead a huge cell with multiple nuclei is formed, making it impossible to test for organellar recycling. What is however apparent is that upon disruption of microtubules, the organisation of the F-actin cytoskeleton is also disrupted, which supports our hypothesis that F-actin and microtubules interact and that this is required for the polarised organisation of parasites. At this stage, we are hesitant to include these data in the manuscript, since they need to be further consolidated and are part of an independent study.

Figure 1. A conditional mutant for alpha-Tubulin demonstrates block in daughter cell development and defects in F-actin dynamics. Parasites co-expressing Cb-EmeraldFP, splitCas9 and a gRNA against *Tub1* were grown in the presence and absence of the inducer Rapamycin (1µM), before immunofluorescence was performed and F-actin and subpellicular microtubules were analysed. Induced parasites demonstrate defects in daughter cell assembly and F-actin dynamics is abrogated and appears to form thick bundles.

2. The authors use actin modulators to affect the dynamics of actin cytoskeleton. This leads to reduction in MIC2 cluster density, especially upon depolymerisation of actin filaments (Fig. 3i). However, despite the seemingly reduced amounts of MIC2, the distribution remains similar to the control (especially at the apical end, Fig 3h). Could the authors please quantify the changes in MIC2 levels and its distribution and comment on the fact that despite actin depolymerisation/stabilisation, MIC2 vesicles still each the apical end? One could speculate that the observed apical signal represents MIC2 that reached the apical end before the cytochalasin D/jasplakinolide treatment. This could be addressed if the authors use the same dual labelling approach they utilised before: they

could label MIC2 with one colour, perform the treatment with actin modulators and then label the newly synthesised MIC2 with another colour to look specifically at the population synthesised/trafficked after the treatment with actin modulators.

We would like to refer the reviewer to our previous publications describing the characterisation of a conditional mutant for TgAct1³⁻⁵. Depletion of actin does not abrogate biogenesis and trafficking of *de novo* synthesised micronemes (or IMC) and their transport appears to require microtubules (see also⁶). The location of *de novo* formed micronemes does not change. In order to test this hypothesis further, we now quantified the ratio of *de novo* versus recycled micronemes in more detail (Fig.2 f,g) and in support of this hypothesis show that recycled micronemes associate preferentially with F-actin.

3. The authors should at least discuss or ideally perform a simple experiment to test whether the actin-dependent mechanism for microneme recycling is specific or whether it reflects a global problem with protein trafficking in the cell in the absence of actin filaments (either because of lack of actin highways for the vesicles or even failure in vesicle formation). The authors could test this by checking what happens to the localisation of several reporter proteins in parasites treated with actin modulators, fixed and stained with specific antibodies.

We performed similar experiments in earlier studies, where we analysed the role of actin in a global manner (see Figure S5 in³ and^{4,5}) using a conditional mutant for *act1*. At this time we did not have a tool available to analyse the distribution of F-actin within the parasite, but analysed the multiple functions of parasite actin during the asexual cycle, including ultrastructural analysis (see⁴, Figure 6F). The only developmental effects we observed previously, is an important role of F-actin in apicoplast inheritance and the formation of parasites with an aberrantly shaped posterior pole, which can be attributed to a recycling defect of the IMC^(5,7) and this study). Another function of actin for transport of dense granules has been found by the Heaslip group⁸. The *de novo* biogenesis of organelles is not significantly affected and using the dual labelling strategy in this study, we show that *de novo* formation of micronemes is actin independent, whereas recycling depends on F-actin. We do speculate that other organelles (as shown here for the IMC), such as rhoptries or dense granules are also recycled in an actin dependent manner and this will be analysed in a future study. Establishment of the dual labelling strategy and performing a similar analysis for other organelles will not be possible within a realistic revision time.

We mention this situation in more detail in the manuscript now.

4. The authors describe differential distribution of the new vs recycled vesicles (Figure 2) and state: "An estimation of the total number of MIC2r and MIC2n indicated that MIC2 vesicles represent two distinct sub-populations: MIC2n can be found mainly at the apical tip of the parasite, while a significant amount of MIC2r appears to be associated with actin bundles localised at the posterior pole (Fig. 2 d,e)." This should be quantified.

We now quantified the ration of MIC2r and MIC2n associated with F-actin (see revised Figure 2). The data fully support our hypothesis that recycled material requires the function of F-actin.

I believe that once the authors provide these additional evidence in support of their hypothesis, the manuscript will be suitable for publishing in Nature Communications and provide a very valuable resource for the scientific community. In particular, the imaging techniques used in this work will

surely inspire other groups and will provide a fantastic guide on how to perform cutting-edge microscopy on intracellular parasites.

We thank the reviewer for the thoughtful comments, which helped us to improve the quality of our study.

Reviewer #2 (Remarks to the Author):

In this manuscript, the authors show the recycling of microneme vesicles from the mother to the daughter cell during *Toxoplasma gondii* replication. Using cutting-edge microscopy, they were able to show that the recycling of vesicle is dependent of actin located inside the parasite as well as in the residual body. The finding is novel and achieved using a technically solid approach, But the overall manuscript needs to be clearer to be understandable for the reader and some points need to be address.

We thank the reviewer for this encouraging and positive assessment of our work.

Major comments

1- What is the ratio of de novo vesicles associated with actin compared to recycled? Are the de novo synthesized vesicles using actin or are they directly made at the apical of the daughter cell?

See comments above. Based on previous research from our and other groups, it appears that *de novo* biogenesis of micronemes occurs along the secretory pathway of the parasite, in analogy to other eukaryotes. The *de novo* formation of the unique secretory organelles has been extensively studied, but to date the recycling pathway we describe here has been unknown.

For a recent review: Venugopal, et al 2018.

We now quantified the ratio of Mic2r and MIC2n associated with F-actin (see comments above).

2- How frequent are the 2 types of vesicular transport dependent of F-actin? Especially the one where the actin self-associate to nearby bundles and form bridge (figure 3e)?

This is indeed a very interesting question. However, when we attempted to quantify the different modes, we reached the limits of our image analysis, since length and diameter of F-actin bundles varies with PV stage. Therefore, we cannot make a definite statement. We now included an illustrative quantification of the functional effect of large F-actin versus short ones and all the ratios of MIC populations (n and r) in a new supplementary figure 2 and include examples of 1,2 and large stage PVs

3- The authors state that actin dynamics is require for IMC recycling, but they show it only for MIC using CD and JAS. Is there any reason why the authors did not check the association of MyoA and actin in presence of CD and JAS?

We would like to refer this reviewer to our previous study, where we demonstrated an IMC defect upon depletion of actin in the parasite, leading to an incomplete IMC (see ⁷). Within this study, the proposed analysis requires clearly defined vesicles we can image. However, recycling of material from the IMC membrane is not a process dependent on well-defined vesicular transport such as the example shown for micronemes. Therefore, the capture and analysis of recycled material and its

quantification with sufficient resolution (live SIM imaging short movies of 1 minute) cannot be as easily performed as for MIC2. Instead, we now provide imaging examples of how material associated to the IMC is transported unambiguously through the F-actin network connecting individual parasites (video S9). We have also added a new supplementary movie of four stage PVs (video S10) treated with CD and JAS that demonstrates that the recycling of vesicles (containing IMC marker MyoA-SNAP) is lost in the presence of CD, arguing for a role of F-actin in recycling material from the IMC.

4- Do the authors have evidence that the actin connects the apicoplast and the trans-golgi network through their data or through literature? In the latter case, a citation should be added and it should not be stated in the legend figure 5e,f,g without data support.

We have corrected this. Indeed, an independent study from the Soldati lab demonstrates polymerisation of F-actin close to the Golgi by Formin 2 (FH-2; see ⁹) and our current study on Formin 2 fully supports this role (see Stortz et al., BioRxiv2018). Both studies have now been discussed in our revision.

5- The authors can't conclude based on their experiment that there is an interplay between the subpellicular microtubules and F-actin in vesicular transport. The authors only show that they are in close proximity and also said in the text that because there is no microtubule in posterior areas of the parasite, it suggests a microtubule independent transport mechanism. They also can't conclude in the legend of figure 5l,m,n that the microtubule connects with actin. Furthermore, what is the conclusion of the long microtubule polymers observed in figure 5h for vesicle transport?

We agree with the reviewer that at this stage the evidences are not fully supporting such an interplay. We toned this hypothesis/interpretation down in our revision. SIM microscopy only improves two-fold the resolution of standard confocal microscope with about 100 nm so we cannot without another set of biochemical experiments conclude that actin and microtubules established a molecular interaction.

Figure 5h together with Fig 5c bottom row (magenta) show findings of novel microtubule dynamics in Apicomplexa resulting in long and short microtubules in close proximity to actin filaments but the SIM based methodology used in these observations cannot conclude a molecular association of actin with microtubules.

Minor comments

1- Figure 1b – what are the number in the picture corresponding to?

We thank the reviewer for raising this question. The number corresponds with the number of estimated recycled vesicles in each cell inside the parasitophorous vacuole. We have rewritten the legend in Fig1 b to clarify this information and it now reads:

b, Top panel. Representative 3D-SIM images show single cells and PVs containing populations of vesicles recycled from the mother cell, and vesicles synthesised de novo after 8h and 18h post invasion event (white arrows show individual cells).

Bottom panel shows the distribution and recycling of vesicles from a single cell stage to two and eight PV stages (estimated numbers in each cell are shown in white)."

2- The author needs to change this sentence to make it clearer. "We also confirmed that micronemes are formed de novo, since after each round of replication the ratio of recycled versus de novo

synthesised MIC2-Halo reflects an even distribution of recycled and de novo synthesised material in individual parasites within a PV (Fig.1b).”

We have now corrected this and it reads now:

“Our observations show that we can distinguish, unambiguously, MIC vesicles formed de novo from those recycled from the mother based on the following evidence. Firstly, the localisation pattern – since, after each round of replication, de novo synthesised MIC2-Halo and recycled vesicles occupy specific locations - de novo, mainly antero apical localisation, and recycled vesicles, redistributed along the cell. Secondly, the recycled vesicles from the mother are distributed evenly in daughter parasites after each replication cycle (Fig.1b).

3- The authors should state in the text that the figure 1c,d,e are on recycling vesicle only as their previous experiments are on both recycled and de novo vesicles.

We have changed it as suggested by the reviewer and now reads:

“We analysed the distribution of MIC2-vesicles being recycled from the mother by looking at single cells and PVs after 6, 12 and 24 h (Fig. 1c,d,e).

4- This sentence is not clear, what does the author wanted to say? Please, reformulate the sentence to make it clearer. “Only a slight non-significant variation occurs after 24h suggesting a level of heterogeneity of maternal microneme vesicles due to processing or different degree of cluster formation (Fig.1e).”

We have corrected this sentence and now it reads:

this variation in the estimated size of the clusters after 24h may suggest a level of heterogeneity of vesicles, either due to processing or a different degree of cluster association which cannot be resolved with the type of fluorophore (TMR) used in the study (Fig.1e).

5- Figure 1h and 3, the unit for the video time should be mention somewhere, for example in the legend.

We have corrected it now and added “(Time scale in frames are shown in minutes)”.

6- Figure 1i, the unit of the transport speed and trajectories should be added in the text when mention. “Pools of recycling vesicles have slightly faster transport (0.267 ± 0.003 versus 0.232 ± 0.004) and larger trajectories (0.421 ± 0.143 versus 0.375 ± 0.16) than de novo synthesised vesicles (Fig. 1i).”

We have corrected these typos and it now reads : “pools of recycled vesicles (two stage 14h PV) have slightly faster transport ($0.267 \pm 0.003 \mu\text{m}\cdot\text{s}^{-1}$ versus $0.232 \pm 0.004 \mu\text{m}\cdot\text{s}^{-1}$) and larger trajectories ($0.421 \pm 0.143 \mu\text{m}\cdot\text{s}^{-1}$ versus $0.375 \pm 0.16 \mu\text{m}\cdot\text{s}^{-1}$) than *de novo* synthesised vesicles (Fig. 1i).”.

7- Figure 1i (actin not labelled) and figure 3f (chromobody), the speed number is considered similar (0.267 and 0.37 to $1.27\mu\text{s}^{-1}$) while the author considered that 0.267 to 0.232 μs^{-1} is different. The same comment applies for the trajectories number. Could the author clarify their statement, maybe with statistical analysis?

We agree with the reviewer that the writing is confusing. We have now rewritten the paragraphs to make the results clearer:

Fig. 1i shows the speed and displacement average of all vesicles in recycled and de novo populations in 14h PVs (two stage vesicles) using manual tracking. Because we averaged the speeds and displacement of recycled and de novo synthesis, this obscures individual events of particles moving very fast. Even using this approach we notice, using a t-test, slight differences in behaviour between recycled and de novo vesicles.

Fig. 3f describes the types of trajectories, localisation and kinetics of individual events using manual tracking, the only method sensitive enough to detect events with very fast displacements in the crowded vesicle environment of the parasite.

We have now re-written the paragraphs to read as follows:

“Quantification of the average speed and trajectories of vesicles using automatic tracking show that pools of recycled vesicles (two stage 14h PV) have slightly faster transport ($0.267 \pm 0.003 \mu\text{m}\cdot\text{s}^{-1}$ versus $0.232 \pm 0.004 \mu\text{m}\cdot\text{s}^{-1}$) and larger trajectories ($0.421 \pm 0.143 \mu\text{m}\cdot\text{s}^{-1}$ versus $0.375 \pm 0.16 \mu\text{m}\cdot\text{s}^{-1}$) than *de novo* synthesised vesicles (Fig. 1i)”.

“Manual tracking enables us to identify variability in the kinetics and trajectories of individual vesicles. MIC2 vesicles exhibited variable speeds (ranging from 0.37 to $1.27 \mu\text{m}\cdot\text{s}^{-1}$) and long net displacement of several microns ($5.4 \mu\text{m}$) which results in exchange of vesicle material between distant daughter cells.”

8- For all the figures, especially for figure 3a, 3i and figure 5, it will be useful to have the apical/basal orientation of the parasite as the pictures are not in the same orientation (keeping the same orientation inside a figure will help the reader).

We have now amended figure 3 and added ‘anterior’ and ‘posterior’ in cartoon 3h together with arrows indicating the direction in which intensity profiles are measured.

9- The word ‘with’ need to be deleted. “In summary, we obtained unambiguous data from fixed images, showing examples of small and big parasitophorous vacuoles, with using live SIM and conventional wide field imaging (Fig.3), we confirmed that MIC2-containing vesicles are associated with a very dynamic F-actin network that connects the posterior end of parasites with the residual body (RB; (Fig. 3a, b, c, Supplementary Fig. 1, Supplementary Video 3-5).”

Now corrected

6- Figure 3 the mic2 vesicles labelled are only recycled or recycled and de novo?

We used a single fluorophore, so the vesicles are all the types of vesicles synthesised until the time of labelling. With the use of one single dye it is not possible to resolve temporally the type of vesicles.

7- Figure 3i, the author should be consistent with the use of abbreviation (CD rather than cyt-D) as well as in the M&M in dual colour SMLM where they used CD D.

We have corrected this inconsistency in the abbreviation and it now reads as CD.

8- Figure 3h the authors said “Right. Intensity profiles of PVs treated with CD show loss of actin network, concentration and re-localisation of MIC2 vesicles to the apical end periphery.” The figure shows the concentration and re-localisation of MIC2 vesicles but not the loss of actin, the authors should reformulate the sentence to say that PV treated with CD where actin network has been loss, show a concentration and re-localisation of MIC2 vesicles.

We have re-written Fig.3h and it now reads: “Right. Intensity profiles of PVs treated with CD show the disappearance of the actin network and a concentration and re-localisation of MIC2 vesicles in the cell periphery.”

9- The author never cited the figure 4h. Furthermore, the colors used in the intensity profiles are difficult to distinguish and there is no mention of their meaning.

We have now linked the text to Fig. 4h and amended the figure legend to clarify the direction of the intensity profile (arrows) and the colour scheme of the intensity profile. The new legend 4h reads as: “h, Intensity profiles from figure G indicate a network of actin (orange line) embedded by MyoA synthesised de novo (MyoAd purple line) and from the mother (MyoA m in cyan line). Intensity was expressed as a percentage of the maximum intensity for each marker. Scale bar 5µm. Z number corresponds to the displayed stack number, brackets show the position of the z stack in the axial plane measured from the parasite surface. MyoA d and MyoA m were imaged using SiR and TMR SNAP ligands respectively. All the images except c, were taken from fixed samples.”

10- Figure 5d and o, the orientation of the profile should be added as it has been done in other figures (ie from a to b).

We have corrected the figure and added arrows showing the direction in which the intensity profile is measured.

11- Figure 6a, the color code is not mention and the time for the video need to be written somewhere.

We have corrected the legend and it now reads:

The experiment was performed using a cell line stably expressing actin-Cb-em (in orange colour) and Myo-SNAP coupled with ligand TMR (in cyan colour)”. We have added a line at the end of the legend fig 6 clarifying the time scale of the movie frames displayed in the figure and reads: “Time in movie frames expressed in minutes”.

12- The schematic model should be better explained in the legend, especially that the unstable actin (due to CD for example) induce clustering of vesicle. The two different actin dependent transport are not clearly described.

We have now redrawn fig 7, divided in three different schematics a,b and c and re-written the figure legend, which now reads:

Fig. 7: a, A single cell translocates MICs from the apical to the posterior end during motility and invasion. After invasion the cell formed a PV containing remaining MIC organelles from the mother (MICr). During endodyogeny replication, the cell is disassembled and maternal MIC2 (MIC2r) accumulates at the RB, which acts as a sorting hub for recycled material, while novel MIC2 (MIC2n) is transported towards the apical tip of daughter parasites. At the late stages of replication, recycled MIC2 is transported from the RB to the apical tip of daughter parasites, where MIC2r and MIC2n form two pools of micronemes. With multiple rounds of replication, the RB becomes more and more complex and expanded, connecting individual parasites within a

PV. This is a general mechanism used in the replication vacuole for long range, multidirectional transport that is guided by a continuous network of F-actin that connects mother, daughter and the recycling body. **b**, F-actin dynamics regulate vesicular transport and cluster formation. We speculate that two populations of MICs can be observed, one formed by recycled vesicles (MICr) that are redistributed using a mechanism depending on F-actin and a second one formed by MICs synthesised de novo and associated with the secretory pathway. Recycled vesicles used F-actin tracks and mobile F-actin for transport and exchange to specific cell locations. **c**, Transported vesicles directly interact with F-actin bundles, facilitating long-range transport and cluster formation. The localisation, cluster density and transport kinetics of the vesicles are controlled by the flow, the level of compaction and mobility of the F-actin scaffold.

13- In the M&M, the author wrote Q80 instead of ku80 (T.gondii transfection and selection section).

We have corrected this

14- For the recycling of MIC2-HALO in replicating PVs, the authors mention that they washed for at least 2h more, what does it mean? They also talk about FOVs but never explain the abbreviation in the manuscript, that likely stand for field of view.

We have now rephrased the text in Experimental procedures section and it now reads:

the dish media were changed three times and left in fresh media for at least 2h. A final change of media was added before imaging to remove dye traces in the solution.

We have now spelled out the field of view acronym (FOV) in figure Fig. 2. Recycled and de novo MIC2 vesicle clusters associate with each other, and bridge actin bundles and in the experimental section Recycling of MIC2-HALO in replicating PVs

15- Supplementary figure1, the legend a is for a movie that is not there, inducing a shift in the following letter.

We have now corrected the labelling of this supplementary figure. Note that with the addition of a supplementary figure to quantify data in Fig.2, it is now supplementary fig.2

16- The author never mentions the time of replication of Toxoplasma (6h), that could help understanding the time point they used in different experiments for reader outside of the field.

We now mention the average replication time of the parasites in the manuscript.

Reviewer #3 (Remarks to the Author):

This work present stunning images and the conclusion of the results on the re-cycling of micronemes and likely other organelles via a F-actin network through the residual body is novel and important for the apicomplexan biology field.

The techniques used based on snaptags, halotags, allowing chasing of micronemes is also novel for Toxoplasma and the images and videos are of amazing quality.

We thank the reviewer for this encouraging and positive assessment of our work.

Critique:

It will be a lot clearer if all images could be separated. For example some of the expanded views of fig 2, b, c, 3b, g, d, e, 4f, 5c, 6b, and others that may be overlapped. Sometimes it is hard to distinguish single from two overlapping images.

We have added a supplementary figure describing the full FOVs of Fig2, which add rendered images of vesicles and a graphical depiction of the estimation number of vesicles in each parasitophorous vacuole which help to clarify the reader the structures described in the Fig.2.

Several figures refer to representative images and one wonders about the number of experiments performed so it will be good to include for each figure the number of experiments performed (n). Please provide statistical analysis when appropriate.

We thank the reviewer for this comment and acknowledge that the term representative images can seem imprecise to the reader. SIM superresolution microscopy is not a high throughput method and statistical analysis is limited in this study. We specified the number of observations in a detailed description in the experimental section and in the annexe for data reproducibility. Our experimental approach was designed to overcome the limitation of a small sample size. We tested the consistency and robustness of our observations using a combination of three complementary microscopy methods, SIM, SMLM, wide-field microscopy, and carried out observations with live and fixed assays, using combinations of dyes.

Is MIC2-Halo secreted normally? Is trafficking altered by the tagging?

This is a very good point and we thank the reviewer for this suggestion. We now include a supplementary figure 5 that includes trails of conclusive secretion of MIC2 de novo and evidence for trails of secreted recycled vesicles.

Additionally, we performed the experiments in three different conditions - stable cell line expressing MIC2-HALO, a stable cell line expressing MIC2-HALO actin chromobody-emerald, and a stable cell line expressing MIC2-HALO transfected transiently with actin chromobody-SNAP. We tested recycling in both stable cell lines and observed recycling of micronemes and vesicle trafficking, concluding that tagging did not block recycling, actin flow in live imaging observations carried out in .

It will be good to include controls for the MyoA-Snap: growth, motility, etc. The cells in Figure 4b look a little swollen.

We have now included controls (see also response above) and an additional supplementary figure 4.

What is the meaning of F-actin bundles encasing the MICr clusters? Wouldn't the clusters move along the network?

We described in Fig 2d and 3d,e and 3i-l how clusters bridge multiple F-actin bundles, possibly resulting in a stabilisation of MIC-actin, and additionally we show that the level of compaction of F-actin Fig 3d-e and Fig3i-l results in changes to localisation and cluster size.

Panel D from fig S1 is missing

Now corrected in the revision

In Fig 3 why does Cytochalasin D and JAS both increase the area considering that they have opposite effects? An explanation would be good.

We summarised our explanations in a new version of Fig 7. Modifying the F-actin structure results in a progressive loss of MIC-actin association, and a displacement of the equilibrium to MIC vesicles associated with each other and the membrane. In CD treatments we think that our observations can be the result of an intermediate transient population of MIC vesicles with short and not fully depolymerised F-actin. Jas treatment could result in transient accumulations of vesicles blocked by the compaction of F-actin.

Figure 4 is confusing about which ones were done with live or fixed cells. Could this be clarified? Does CD and Jas affect MyoA traffic also?

We provide additional movies that suggest that the recycling and trafficking of IMC material can be affected by the changes in actin structure (supplementary video 10).

A sporozoite is mentioned in Figure 5, should it be tachyzoite?

It is corrected now.

Could the authors provide cartoons showing the strategies for tagging in the supplemental material? Why there is not a separate discussion?

A schematic of the constructs and strains are shown in Supplementary fig.6 in the experimental section

We now added a separate discussion.

References

- 1 Nichols, B. A., Chiappino, M. L. & O'Connor, G. R. Secretion from the rhoptries of *Toxoplasma gondii* during host-cell invasion. *J Ultrastruct Res* **83**, 85-98 (1983).
- 2 Morrisette, N. S., Murray, J. M. & Roos, D. S. Subpellicular microtubules associate with an intramembranous particle lattice in the protozoan parasite *Toxoplasma gondii*. *J Cell Sci* **110 (Pt 1)**, 35-42 (1997).
- 3 Andenmatten, N. *et al.* Conditional genome engineering in *Toxoplasma gondii* uncovers alternative invasion mechanisms. *Nat Methods* **10**, 125-127, doi:10.1038/nmeth.2301 (2013).
- 4 Egarter, S. *et al.* The toxoplasma Acto-MyoA motor complex is important but not essential for gliding motility and host cell invasion. *PLoS One* **9**, e91819, doi:10.1371/journal.pone.0091819 (2014).
- 5 Whitelaw, J. A. *et al.* Surface attachment, promoted by the actomyosin system of *Toxoplasma gondii* is important for efficient gliding motility and invasion. *BMC Biol* **15**, 1, doi:10.1186/s12915-016-0343-5 (2017).
- 6 Shaw, M. K., Compton, H. L., Roos, D. S. & Tilney, L. G. Microtubules, but not actin filaments, drive daughter cell budding and cell division in *Toxoplasma gondii*. *J Cell Sci* **113 (Pt 7)**, 1241-1254 (2000).

- 7 Periz, J. *et al.* Toxoplasma gondii F-actin forms an extensive filamentous network required for material exchange and parasite maturation. *Elife* **6** (2017).
- 8 Heaslip, A. T., Nelson, S. R. & Warshaw, D. M. Dense granule trafficking in Toxoplasma gondii requires a unique class 27 myosin and actin filaments. *Mol Biol Cell* **27**, 2080-2089 (2016).
- 9 Tosetti, N., Dos Santos Pacheco, N., Soldati-Favre, D. & Jacot, D. Three F-actin assembly centers regulate organelle inheritance, cell-cell communication and motility in Toxoplasma gondii. *Elife* **8**, doi:10.7554/eLife.42669 (2019).

Reviewers' Comments:

Reviewer #1:

Remarks to the Author:

The authors have successfully clarified the manuscript. In particular, they included additional quantifications and explanations for the statistics used, included additional controls and toned down their conclusions. These changes improved the manuscript while maintaining the original significance of the work and proving its high quality. I believe that this elegant study is suitable for publication in Nature Communications and will provide a great advancement in the field.

Michal Pasternak

Reviewer #3:

Remarks to the Author:

The authors did a great job in responding to the critiques and concerns of this reviewer.

Reviewer #4:

Remarks to the Author:

The authors made a lot of effort to improve the manuscript and make it clearer for the reader. They have address all of our questions with serious and overall the manuscript is of high quality. This is a really well done and novel work proposed here.